# BlockSpec: Blockwise Speculative Decoding for Diffusion LLMs

## Abstract

In diffusion-based Large Language Models (dLLMs), parallel decoding is usually realized through threshold-based or top-k strategies. While effective in high-confidence tokens, these strategies often collapse on low-confidence tokens, forcing the model into inefficient single-token decoding. To address this limitation, we propose Block Speculation (BlockSpec), a novel training-free blockwise speculative decoding method that explores multiple future decoding trajectories in parallel. Our method introduces a new tree-based trajectory generation strategy and a blockwise parallel verification module, where decoding tokens are organized into tree exploration paths and then multiple decoding trajectories can be simultaneously verified. Unlike traditional speculative decoding that focuses only on fixed-order left-to-right token speculation, our approach introduce the block-level speculation, which jointly explores both token choices and decoding trajectories for dLLMs. We also design two complementary speculation formulations—intra-block and inter-block speculation—that jointly accelerate dLLMs within and across blocks. Extensive experiments show that the proposed BlockSpec model reduces iteration steps by up to 40%, accelerating over 80% of decoding steps. As a result, our model achieves up to 7–14× speedup over vanilla dLLMs, together with an additional 1.3× improvement over state-of-the-art methods.

## 1 Introduction

Diffusion-based Large Language Models (dLLMs) (Nie et al., 2025; Ye et al., 2025; Song et al., 2025; Khanna et al., 2025) have recently emerged as a compelling alternative to traditional autoregressive language models, drawing increasing attention for their potential to deliver more efficient inference. In contrast to the autoregressive paradigm, where tokens are generated sequentially from left to right, dLLMs formulate text generation as a parallel denoising process.

Masked Diffusion Language Models (MDLMs) (Sahoo et al., 2024; Zheng et al., 2023) stand out as the most widely adopted variant among dLLMs. MDLMs frame the text generation as a denoising process that begins with a sequence of [MASK] tokens and gradually refines them into concrete tokens. In practice, parallel decoding is often performed by selecting tokens with the highest confidence or those that exceed a predefined threshold for simultaneous decoding. However, many studies have shown that simply increasing the level of parallelism leads to a significant drop in model accuracy, revealing an inherent trade-off between decoding efficiency and generation quality for dLLMs.

Through empirical analysis of parallel decoding strategies in state-of-the-art dLLMs, we observe a recurring limitation in existing strategies. While these parallel decoding methods can unmask multiple tokens per step, this ability breaks down when the model encounters low-confidence tokens. To preserve prediction accuracy, the decoder must degenerate to decode only a single token with highest confidence at that step. We refer to this degeneration from multi-token per step to single-token decoding under low-confidence conditions as low-confidence degeneration. Using Fast-dLLM (Wu et al., 2025) as our experimental parallel decoding strategies, we find that such low-confidence degeneration events occur frequently in practice (Figure 1), forming a critical bottleneck that limits the efficiency of diffusion-based parallel decoding.

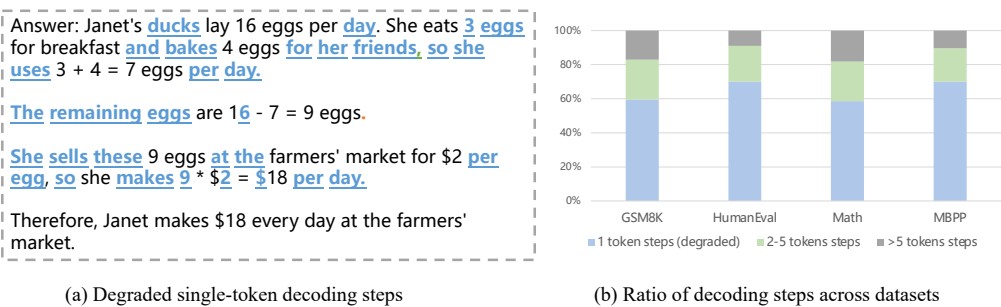

(a) Degraded single-token decoding steps  (b) Ratio of decoding steps across datasets

Figure 1: Visualization of low-confidence degeneration issues of parallel decoding in dLLMs. (a) The degeneration steps in which model only decode single top confidence token. (b) Ratio of degraded decoding steps across datasets.

Our analysis reveals that low-confidence degeneration is a pervasive issue across diverse datasets—whenever it occurs, the model's decoding efficiency drops sharply.We observe that such low-confidence tokens may arise from either multiple plausible continuation candidates (Kim et al., 2025)or inherently difficult reasoning steps. Prior methods typically treat these tokens as unreliable and simply discard or avoid them. In contrast, we argue that low-confidence tokens can still be valuable, as uncertainty does not necessarily imply incorrectness. This motivates our core assumption: even when confidence is low, the correct continuation may still be among the candidate tokens, and parallel verification provides an effective way to retain these possibilities without sacrificing efficiency. Building on this idea, we introduce Block Speculation, a training-free speculative decoding framework that performs blockwise lookahead to substantially improve decoding efficiency for diffusion-based language models.

Speculative decoding has been widely adopted in autoregressive language models as an effective approach to accelerate generation by exploring multiple upcoming tokens in parallel. Its success largely relies on two key properties: fast draft generation and efficient one-step parallel verification. However, these speculative decoding methods cannot be simply applied to diffusion-based language models due to dLLMs' design of any-order decoding and bidirectional attention.

The first core component of our Block Speculation is the draft generation strategy. In autoregressive models, draft generation simply follows a fixed left-to-right decoding order and only needs to select the most likely next tokens. In contrast, draft generation in diffusion language models must construct an entire denoising/decoding trajectory, which not only selects candidate tokens but also determines their positions and ordering. To address this, we propose a tree-based decoding trajectory construction strategy, where each node corresponds to a complete block of tokens. The tree root represents the initial state, and the tree paths from root enable the exploration of multiple alternative denoising trajectories in parallel. Unlike the token-by-token draft generation in autoregressive models, Our strategy enables the generation of decoding trajectories with arbitrary token orders and arbitrary degrees of parallelism, which fully capturing the denoising process of dLLMs.

The second core component of our method is the block-level parallel verification mechanism. In autoregressive models, draft token sequences can be verified in a single step using a causal attention mask. However, this token-level verification is infeasible for diffusion language models due to their bidirectional attention design. In our model, we construct a blockwise semi-autoregressive attention masks, which enable parallel verification across different nodes of the draft tree. During verification, we sequentially traverse each path in the draft tree, checking the correctness of its denoising trajectories. Our model introduces additional draft block tokens for verification, which increases the computational cost by approximately $K \times$ blocksize, where $K$ is the number of draft block nodes. We benchmarked this extra overhead across different GPU architectures and figured

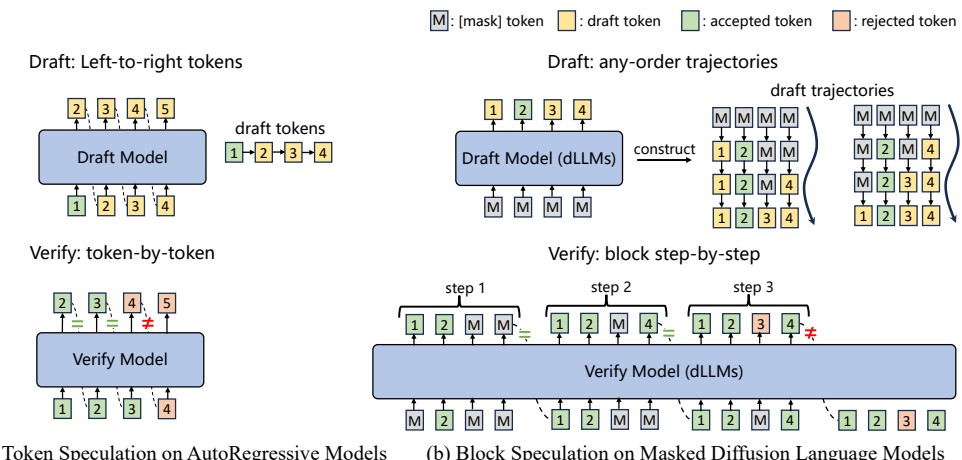

(a) Token Speculation on AutoRegressive Models     (b) Block Speculation on Masked Diffusion Language Models

Figure 2: Visualization of vanilla token speculation in AutoRegressive(AR) models and our block speculation in masked diffusion language models (dLLMs). M denotes a `[MASK]` token. Numbers k (e.g., 1) correspond to token at position k. The block size is 4 here.

out the most efficient structure. Figure2 shows the difference of common token speculation methods with our proposed block speculation.

We further design our framework around a semi-autoregressive decoding paradigm as Block diffusion(Arriola et al., 2025), where the dLLM model generates text block by block in a left-to-right manner, while performing any-order diffusion decoding within each block. Under this paradigm, Block Speculation enables complementary intra-block and inter-block forms of speculative decoding: it can perform parallel token speculation within a single block, as well as parallel block speculation across successive blocks. This dual-level design further enhances the parallel efficiency of speculative decoding in dLLMs.

Our main contributions can be summarized as follows:

- We identify the low-confidence degeneration problem as a fundamental limitation of parallel decoding in dLLMs and introduce a novel Block Speculation framework as an effective solution.

- We propose a novel tree-based trajectory generation strategy together with a blockwise semi-autoregressive verification module, which are specifically tailored to the any-order decoding paradigm and bidirectional attention design of dLLMs.

- We design two complementary speculation decoding formulations — intra-block speculation and inter-block speculation—which together enhance decoding efficiency across both local and cross-block levels. Extensive experiments demonstrate that our approach reduces iteration steps by up to 40%, mitigates over 80% of low-confidence decoding cases, and achieves up to $14\times$ speedup over vanilla dLLMs, with an additional $1.3\times$ improvement over state-of-the-art methods.

## 2 RELATED WORK

### 2.1 PRELIMINARY OF DIFFUSION LARGE LANGUAGE LODELS

Diffusion-based language models (dLLMs) have recently gained significant attention as an emerging paradigm for large-scale language modeling, with representative efforts including open-source models such as LLaDA(Nie et al., 2025) and Dream(Ye et al., 2025), as well as closed-source systems like Seed Diffusion(Song et al., 2025) and Mercury(Khanna et al., 2025). Unlike traditional autoregressive models that rely on a left-to-right next-token prediction paradigm, dLLMs reformulate text generation as a denoising process applied to the entire sequence.

A common formulation is the masked diffusion language model (MDLM)(Sahoo et al., 2024), which introduces noise into the sequence through iterative forward masking and then learns to reconstruct the original sequence via a reverse denoising process. Given a token sequence $\mathbf{x}_0 = (x_1, x_2, \ldots, x_n)$, the forward process gradually replaces tokens with a special mask token [MASK] according to a corruption rate $\beta_t$:

$$q(\mathbf{x}_t \mid \mathbf{x}_{t-1}) = \prod_{i=1}^{n} \Big[ (1 - \beta_t)\,\delta(x_{t,i} = x_{t-1,i}) + \beta_t\,\delta(x_{t,i} = [\text{MASK}]) \Big], \qquad (1)$$

which after $T$ steps yields a fully masked sequence $\mathbf{x}_T$. The reverse denoise process is parameterized by the diffusion model $p_\theta$, which progressively denoises masked sequences back to valid tokens.

## 2.2 PARALLEL DECODING IN DLLMS

Parallel decoding has been widely explored to accelerate inference in diffusion language models (dLLMs). Two representative strategies are top-$k$ based decoding and threshold-based decoding.

**Top-$k$ decoding.** As adopted in LLaDA (Nie et al., 2025), the model selects the k most confident tokens at each step for denoising in parallel. While this guarantees a fixed degree of parallelism, it often becomes unreliable in low-confidence regimes, where the selected tokens are likely to include incorrect predictions that degrade accuracy.

**Threshold-based decoding.** Fast-dLLM (Wu et al., 2025) instead selects all tokens whose confidence scores exceed a predefined threshold $\tau$. This strategy adaptively adjusts the parallel width according to model confidence, offering more flexibility than a fixed $k$. However, when no token surpasses the threshold, it collapses to top-1 decoding, resulting in severe low-confidence degeneration.

Our approach leverages blockwise draft generation and parallel verification, which sustains parallelism without sacrificing accuracy, even under low-confidence conditions where prior methods fail.

## 2.3 SPECULATIVE DECODING

**Speculative decoding in Autoregressive Models.** Speculative decoding has emerged as an effective technique for accelerating autoregressive language model inference (Chen et al., 2023; Leviathan et al., 2023). These approaches follow a draft–verify workflow: a smaller or faster draft model first proposes candidate tokens, and the target model then verifies them in parallel under a causal attention mask, accepting valid tokens and rejecting others. This mechanism enables the decoding of multiple tokens within a single inference step. Building on this idea, subsequent efforts have expanded both draft-generation strategies and verification policies. Multi-model designs (Xia et al., 2022; Miao et al., 2024) leverage lightweight draft models, whereas single-model or self-speculative methods (Cai et al., 2024; Li et al., 2024) rely on the target model itself to construct drafts efficiently. Recent work further investigates how to optimize draft construction and alignment for stronger acceptance behavior. Goel et al. (2024) propose directly aligning draft models to chat-tuned LLMs, improving consistency and reducing verification failures. In parallel, Jeon et al. (2024) introduce recursive speculative decoding, which employs sampling without replacement to iteratively refine draft candidates. Together, these developments highlight the growing interest in more robust and efficient speculative decoding frameworks.

**Speculative decoding in dLLMs.** Recent studies have also begun applying speculative decoding to diffusion-based language models (dLLMs). Adaptive Parallel Decoding (Israel et al., 2025) adopts a dual-model scheme where a small autoregressive model serves as a verifier. Two very recent works (De Bortoli et al., 2025; Hu et al., 2025) explore speculative sampling for continuous denoising diffusion models, enabling parallelization in DDPM-style generative processes. Concurrently, Agrawal et al. (2025) propose a block-level speculative decoding framework. While conceptually related, these approaches focus on denoising diffusion or rely on complex calibration mechanisms, whereas our work targets discrete mask–based diffusion LLMs and is grounded in an analysis of the low-confidence degeneration problem. This leads us to prioritize candidates that remain reliable after confidence decay and to adopt a lightweight tree-based draft structure, further enhanced by inter-block speculation. These design choices allow our method to achieve substantially higher

speed—approximately $14\times$ compared to Spiffy's $7\times$ under similar settings. Additional comparisons with Agrawal et al. (2025) are provided in Appendix A.4.

## 3 METHODOLOGY

### 3.1 GENERAL FRAMEWORK

Our overall framework is illustrated in Figure 3 and consists of four major steps: Step 1,2 for draft generation and Step 3,4 for parallel verification.

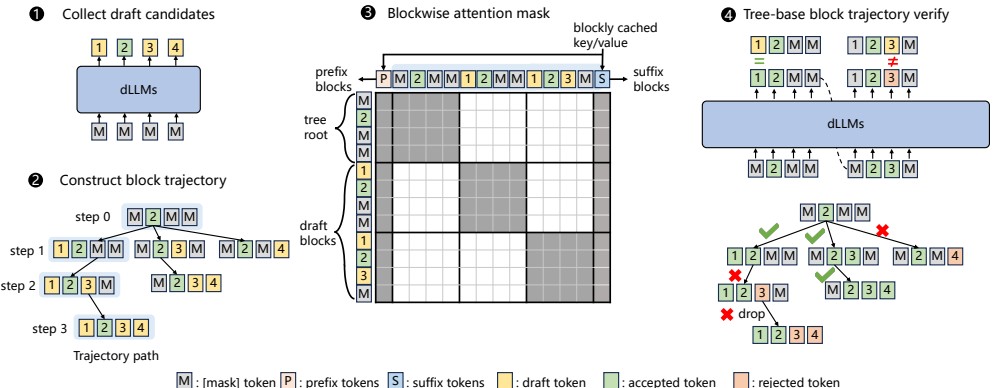

Figure 3: General framework of Block Speculation. Numbers k (e.g., 1) correspond to token at position k. Number 0 refers the prompt tokens. The block size is $4$ here. The prefix and suffix tokens' KV states are refreshed at the beginning of each block via a full-length full-attention forward pass and then cached for blockwise reuse.

In Step 1, possible draft tokens are gathered based on historical predictions from the base dLLM. Since this step directly reuses the original dLLM without external draft models, our method belongs to the family of self-speculative decoding. In Step 2, the collected candidate tokens are prioritized by confidence and incrementally assembled into multiple decoding paths starting from the initial state, where each path corresponds to a branch in the tree-based block trajectory.

Once the tree-based draft blocks are constructed (Step 2), they are concatenated with the input sequence and evaluated jointly (Step 3). To preserve the bidirectional attention structure of dLLMs while preventing interference across draft branches, we apply a blockwise individual attention mask: within each draft block, all tokens are visible to the blockly-cached prefix and suffix states, but draft tokens from different blocks remain mutually invisible. In Step 4, all draft blocks are verified in parallel: the model's predictions are compared with each draft trajectory independently, and the acceptance strategy selects the valid decoding path for the final output.

Our overall framework involves two dLLM forward passes: the first for draft trajectory generation and the second for parallel verification. In practice, these two passes can be merged into a single shared forward across consecutive block speculation steps, allowing the model to simultaneously perform verification for the current step and draft generation for the next. This design further reduces general model's inference overhead.

### 3.2 DRAFT TRAJECTORY GENERATION

Our draft trajectory generation module consists of candidate collection and trajectory construction. The core idea is that, even when a position is classified as low-confidence, it may still contain reasonable candidate tokens. Prior methods, however, typically discard these candidates once their confidence falls below the threshold $\tau$. To better leverage this signal, we collect the top-$k$ candidates at low-confidence positions—that is, tokens that appear within the top-$k$ probability ranks but whose confidence is lower than $\tau$. Token confidence is defined as

$$c(x) = \max(\text{softmax}(\text{logits}_t(x))). \tag{2}$$

We then construct draft trajectories in a tree-based, top-down manner. At the root layer, the top-$k$ candidates initialize the tree nodes, maximizing the likelihood of including correct predictions early on. Each node could recursively expanded by appending additional top-$k$ tokens not yet present in the path. In practice, expanding more nodes generally increases trajectory coverage but also raises the computational overhead during verification. To obtain a good balance between accuracy and efficiency, we adopt a hybrid node expansion strategy: the root node expands with top-$k$ candidates, while all deeper layers expand only the top-1 candidate for each active node. This keeps the tree compact while preserving sufficient exploratory breadth at the root, yielding the best trade-off observed in our experiments. (More details in Appendix A.2.)

The number of nodes in the constructed trajectory tree is bounded by $O(D \cdot W)$, where $D$ is the maximum depth and $W$ the maximum width. In practice, the tree is rarely complete: construction is biased toward expanding higher-confidence nodes to deeper levels, reflecting the assumption that the joint probability mass is concentrated along high-confidence paths. For notation, we denote a tree as $W3D3(6)$, meaning $max\_width = 3$, $max\_depth = 3$ with a total of 6 children nodes (as the example in Figure3).

We also experimented with alternative draft generation strategies, including threshold-based tree construction. However, as shown in Section 4.3 these variants did not yield significant improvements in draft accuracy or token recall. Therefore, our final model adopt the concise and effective Top-$k$ version. More example tree structures and generation strategies are presented in Appendix.

---

**Algorithm 1:** Tree-based Draft Trajectory Generation

---

**Input:** Previous model logits $\{\ell_t\}$, parameters $k$, $D$, $W$, threshold $\tau$
**Output:** Draft trajectory tree $\mathcal{T}$
Initialize $\mathcal{T}$ with root node $T_0$ and an empty token set $\text{tokens}(T_0)$ ;
**foreach** *token* $x$ **do**
    $c(x) \leftarrow \max(\text{softmax}(\ell_t(x)))$ ;
$S_{\text{high}} \leftarrow \{\, x \mid c(x) > \tau \,\}$;
$\text{tokens}(T_0) \leftarrow \text{tokens}(T_0) \cup S_{\text{high}}$ ;
Candidate pool: $C \leftarrow \{\, x \mid x \notin \text{tokens}(T_0) \,\}$ ;
**for** $\ell = 1$ **to** $D$ **do**
    **if** $\ell = 1$ **then**
        $S \leftarrow$ top-$k$ tokens from $C$ ranked by $c(x)$ ;
        **foreach** $x \in S$ **do**
            create child node $n$ of $T_0$ with $\text{tokens}(n) \leftarrow \text{tokens}(T_0) \cup \{x\}$ ;
    **else**
        **foreach** *node $n$ at layer $\ell - 1$* **do**
            $S \leftarrow$ top-$k$ tokens from $\big(C \setminus \text{tokens}(n)\big)$ ranked by $c(x)$ ;
            **foreach** $x \in S$ **do**
                create child node $m$ of $n$ with $\text{tokens}(m) \leftarrow \text{tokens}(n) \cup \{x\}$ ;
    prune nodes of layer $\ell$ to width $W$ ;
**return** $\mathcal{T}$

---

### 3.3 BLOCK PARALLEL VERIFICATION

After draft blocks are generated, our parallel verification module evaluates them using a blockwise speculative mechanism. Unlike autoregressive speculative decoding, token-level causal masking cannot be directly applied to dLLMs because denoising relies on [MASK] tokens. To address this, we adopt a blockwise individual attention scheme: within each draft block, every draft token is visible to all blockly-cached prefix and suffix tokens, while draft tokens from different branches remain mutually invisible to prevent cross-branch interference. The kv cache desgion aligns with Fast-

dLLM (Dual Cache) as detailed in Sec 3.4. This attention mask strategy preserves the bidirectional structure of dLLMs while enabling independent draft expansion, as shown in Figure 3.

During verification, all draft blocks are concatenated with the input and processed in a single forward pass. The predictions are matched to the draft trajectory tree for path-by-path verification: each block is accepted only if its tokens appear in the top-$k$ predictions or exceed a confidence threshold; otherwise, the block and its descendants are pruned. This selective verification preserves efficiency while maintaining reliability. Because draft generation and verification share the same forward pass, the additional inference overhead is minimal.

### 3.4 DESIGN OF SUFFIX TOKENS AND KV CACHE

**Retention of Suffix Tokens.** In masked diffusion language models (MDLMs), decoding begins from a fully masked sequence of length $L$, and the bidirectional attention used during training requires all unresolved suffix positions to remain visible during inference. We empirically find that removing or truncating suffix [MASK] tokens leads to a substantial drop in accuracy, as it creates a severe train–inference attention misalignment. For this reason, our method preserves the entire suffix throughout all decoding steps, and accelerates their processing using the KV Cache mechanism described below.

**KV Cache Mechanism.** Previous works like Fast-dLLM(Dual Cache) (Wu et al., 2025; Chen et al., 2025) show that KV activations of prefix and suffix tokens remain highly similar across adjacent denoising steps in dLLMs, and reusing them within a block leads to only a minor ($\sim$1-point) accuracy drop. Building on this observation, BlockSpec adopts a similar dual-cache design with Fast-dLLM(Dual Cache): at the start of each block's speculative decoding, we compute the complete KV states for the prefix (prompt and decoded tokens), the suffix (remaining [MASK] tokens), and the current block root. The prefix and suffix KV states are cached and remain fixed throughout the subsequent block decoding steps. They are reused for all draft expansions and verification passes, while only the KV entries associated with actively explored draft tokens are updated.

To distinguish this mechanism from traditional causal KV caches in autoregressive decoding, we explicitly refer to these reused out-of-block KV states (prefix and suffix) as blockly-cached KV. This terminology emphasizes that these KV states are updated only at block boundaries and then cached within a block, enabling substantial computational savings while preserving the bidirectional attention structure of diffusion LLMs.

### 3.5 INTER-BLOCK SPECULATION

While intra-block speculation substantially improves local efficiency, it remains confined to the current block.

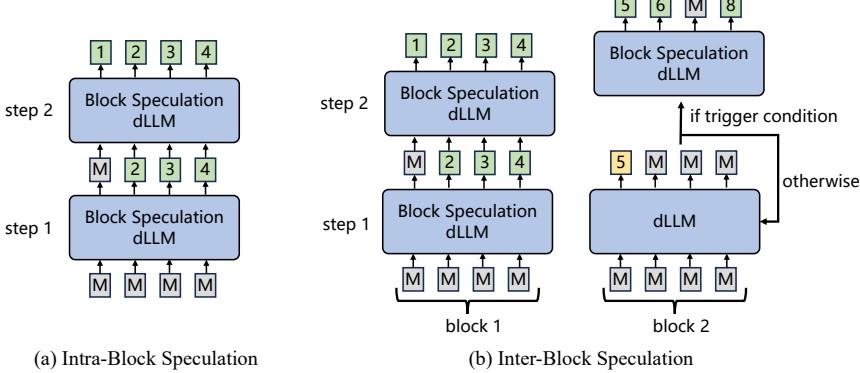

(a) Intra-Block Speculation      (b) Inter-Block Speculation

Figure 4: Visualization of Intra-block Speculation and Inter-block Speculation.

The inter-block attention mask follows a similar blockwise formulation while preserving the bidirectional structure of of inter-block roots. In our implementation, the roots of the two adjacent blocks

retain full bidirectional visibility: all draft tokens of the next block are visible to the current block's root, and all draft tokens of the current block are likewise visible to the next block's root. Thus, the two blocks share each other's root Key/Value states, consistent with prior observations in Fast-dLLM that KV activations of prefix and suffix tokens remain highly similar across adjacent denoising steps. Draft tokens across different blocks remain mutually invisible, ensuring that intra-block exploration is isolated. In practice, inter-block speculation is applied only to two consecutive blocks, and their draft trajectories are generated independently. Additional details and visualizations of the inter-block attention mask are provided in Appendix A.3.

## 4 EXPERIMENTS AND ANALYSIS

### 4.1 EXPERIMENTAL CONFIGURATIONS AND REPRODUCIBILITY

We evaluate our approach on two diffusion-based LLMs, LLaDA-Instruct-7B(Nie et al., 2025) and Dream-Instruct-7B(Ye et al., 2025). We employ the Fast-dLLM(dual-cache) as baseline models. The dual kvcache mechanism of Fast-dLLM (Dual Cache) reduces redundant recomputation and improves throughput. Unless otherwise specified, all our comparisons and proposed methods are implemented based on Fast-dLLM (Dual Cache) which means that we cached the prefix and suffix key-values. Experiments are conducted on mathematical reasoning (GSM8K, MATH) and code generation (HumanEval, MBPP) benchmarks. For fairness, we fix the generation length to 512 tokens and the blocksize to 32. Results of different generation lengths are provided in Appendix A.9. We mostly use the W2D2(3) tree structure as we analysis in Sec 4.3. While a larger structure such as W3D3(6) yields a higher acceptance rate and tokens-per-step, we empirically observed that W2D2(3) provides the best overall speed–efficiency trade-off On A800. On higher-end hardware (e.g., H800), W3D3(6) performs better, but for fairness with the baseline implementations, most main results are reported using the A800 W2D2(3) configuration. For better reproducibility, we have included the detailed settings and implementation in the AppendixA.1. The source code and reproducible environment dockers will be published soon.

### 4.2 GENERAL RESULTS

The general performance of our method on LLaDA and Dream is summarized in Table 1 and Table 2. Across all evaluated datasets, Block Speculation consistently achieves substantial reductions in the number of decoding steps. Specifically, the average decoding length can be reduced from the original 512 steps to less than 120 steps, representing a significant compression of the denoising trajectory. Notably, our method also outperforms the parallel decoding baseline Fast-dLLM, reducing its step count by an additional 43%.

| Benchmark | Method | TPS↑ | Latency (s)↓ | Gen. Length | Steps↓ | Score↑ | TPS Speedup↑ |
|---|---|---|---|---|---|---|---|
| GSM8K(4) | LLaDA-Instruct | 4.1 | 65.6 | 267.8 | 512.0 | 76.6 | 1.0× |
| | Fast-dLLM | 21.3 | 12.6 | 268.4 | 95.3 | 77.3 | 5.2× |
| | Fast-dLLM (Dual Cache) | 48.8 | 5.4 | 263.7 | 110.3 | 75.6 | 11.9× |
| | BlockSpec (Ours) | 55.7 | 4.8 | 266.5 | 71.3 | 75.5 | **13.6×** |
| HumanEval(0) | LLaDA-Instruct | 13.8 | 34.4 | 474.0 | 512.0 | 43.9 | 1.0× |
| | Fast-dLLM | 41.5 | 11.5 | 476.3 | 163.6 | 43.3 | 3.0× |
| | Fast-dLLM (Dual Cache) | 70.6 | 6.6 | 467.0 | 179.2 | 45.7 | 5.1× |
| | BlockSpec (Ours) | 94.4 | 5.0 | 467.3 | 105.1 | 46.3 | **6.8×** |
| MATH(4) | LLaDA-Instruct | 7.2 | 59.4 | 430.2 | 512.0 | 36.8 | 1.0× |
| | Fast-dLLM | 23.7 | 18.1 | 430.2 | 148.0 | 36.8 | 3.3× |
| | Fast-dLLM (Dual Cache) | 60.7 | 7.1 | 428.3 | 167.1 | 35.7 | 8.4× |
| | BlockSpec (Ours) | 73.5 | 5.8 | 428.7 | 96.8 | 36.0 | **10.2×** |
| MBPP(3) | LLaDA-Instruct | 4.8 | 61.6 | 298.2 | 512.0 | 14.8 | 1.0× |
| | Fast-dLLM | 20.8 | 14.4 | 299.0 | 115.2 | 15.0 | 4.3× |
| | Fast-dLLM (Dual Cache) | 48.5 | 6.0 | 293.0 | 132.8 | 13.6 | 10.1× |
| | BlockSpec (Ours) | 56.4 | 5.1 | 285.8 | 81.7 | 13.4 | **11.8×** |

Table 1: Performance comparison of different dLLM methods based on LLaDA-Instruct-7B. Number in parentheses indicates the number of shots. The generation length is fixed to 512.

| Benchmark | Method | TPS↑ | Latency (s)↓ | Gen. Length | Steps↓ | Score↑ | TPS Speedup↑ |
|-----------|--------|------|--------------|-------------|--------|--------|--------------|
| GSM8K(4) | Dream-base | 9.0 | 56.6 | 511.0 | 512.0 | 76.8 | 1.0× |
|  | Fast-dLLM | 16.3 | 31.4 | 511.0 | 252.6 | 74.5 | 1.8× |
|  | Fast-dLLM (Dual Cache) | 50.6 | 10.1 | 510.7 | 280.6 | 74.6 | 5.6× |
|  | BlockSpec (Ours) | 62.2 | 8.2 | 510.8 | 164.9 | 73.7 | **6.9×** |
| HumanEval(0) | Dream-base | 16.9 | 30.2 | 511.0 | 512.0 | 54.2 | 1.0× |
|  | Fast-dLLM | 28.6 | 17.9 | 511.0 | 247.3 | 51.8 | 1.7× |
|  | Fast-dLLM (Dual Cache) | 52.3 | 9.8 | 510.9 | 295.0 | 51.2 | 3.1× |
|  | BlockSpec (Ours) | 78.1 | 6.5 | 510.9 | 180.7 | 54.3 | **4.6×** |
| MATH(4) | Dream-base | 9.8 | 52.2 | 512.0 | 512.0 | 39.7 | 1.0× |
|  | Fast-dLLM | 32.5 | 15.7 | 512.0 | 133.6 | 39.5 | 3.3× |
|  | Fast-dLLM (Dual Cache) | 74.9 | 6.8 | 511.9 | 181.7 | 38.0 | 7.6× |
|  | BlockSpec (Ours) | 103.4 | 5.0 | 511.9 | 115.5 | 38.0 | **10.6×** |
| MBPP(3) | Dream-base | 10.0 | 51.0 | 512.0 | 512.0 | 56.8 | 1.0× |
|  | Fast-dLLM | 37.1 | 13.8 | 512.0 | 116.7 | 55.4 | 3.7× |
|  | Fast-dLLM (Dual Cache) | 84.3 | 6.1 | 512.0 | 157.5 | 55.2 | 8.4× |
|  | BlockSpec (Ours) | 108.7 | 4.7 | 512.0 | 87.3 | 54.4 | **10.9×** |

Table 2: Performance comparison of different dLLM methods based on Dream-base models. Number in parentheses indicates the number of shots. The generation length is fixed to 512.

In terms of runtime efficiency, our method yields consistent improvements in tokens per second (TPS). Compared with Fast-dLLM, Block Speculation achieves a speedup of approximately 15%–35% across different datasets. The slight ($\sim$1%) accuracy differences relative to Fast-dLLM (Dual Cache) arise from trajectory shifts under higher parallelism and blockwise discrepancies introduced by inter-block interactions. These results demonstrate that our approach not only alleviates the low-confidence degeneration problem but also establishes a more efficient decoding paradigm for diffusion-based language models.

### 4.3 ANALYSIS OF DRAFT GENERATION AND INTERBLOCK

The draft generation strategy constitutes one of the most critical components of our framework, as both the accuracy and acceptance rate of generated drafts directly determine the overall speedup achievable by Block Speculation. To better understand its impact, we conduct an ablation study that systematically evaluates contribution of the tree structure and draft collect methods in Table3.

| Methods | Tree Configs | Step1 AR | Step2 AR | Step3 AR | Tokens/Step | Steps/Answer | Latency (s) |
|---------|--------------|----------|----------|----------|-------------|--------------|-------------|
| **Baseline** Fast-dLLM (Dual Cache) | – | – | – | – | 2.61 | 179.2 | 6.6 |
| **+ BlockSpec** |  |  |  |  |  |  |  |
|  | W1D1(1) | 0.62 | – | – | 3.51 | 134.5 | 5.6 |
|  | W1D3(3) | 0.69 | 0.65 | 0.49 | 3.66 | 129.8 | 5.4 |
|  | W3D1(3) | 0.78 | – | – | 3.82 | 123.3 | 5.4 |
|  | W2D2(3) | 0.78 | 0.56 | – | 3.83 | 124.0 | **5.3** |
|  | W3D3(6) | 0.83 | 0.59 | 0.43 | **3.95** | **120.1** | 5.5 |
| **+ BlockSpec + Interblock** |  |  |  |  |  |  |  |
|  | W2D2(3) | 0.78 | 0.57 | – | 4.28 | 106.7 | **5.0** |
|  | W3D3(6) | 0.82 | 0.59 | 0.43 | **4.52** | **105.1** | 5.6 |

Table 3: Ablation study showing incremental improvements from the baseline fastdllm (dual cache), to BlockSpec, and to BlockSpec + Interblock on A800. Results are reported on LLaDA-7B-Instruct with generation length 512 on Humaneval. Here, AR denotes the acceptance rate at specific draft steps . While the W3D3 structure achieves the highest tokens/step and the fewest decoding steps, its practical latency is noticeably worse than W2D2 due to heavier per-step verification and draft-generation overhead which motivates evaluating on higher-compute hardware to better reveal its potential.

We further study the effect of inter-block speculation in Table 3. Within the tree-based trajectory exploration framework, adding inter-block speculation reduces total decoding iterations by about 12% and significantly increases the number of verified tokens per step, leading to better synchronization efficiency. Although acceptance rates remain largely unchanged, the higher per-step throughput explains the overall speed gains. While W3D3 attains the highest tokens-per-step and the fewest steps,

its per-step verification cost results in higher latency than W2D2, suggesting that its advantages may be more pronounced on stronger hardware like H800.

## 4.4 ANALYSIS OF COMPUTING OVERHEAD

Our method mitigates low-confidence degeneration by exploring multiple blockwise trajectories in parallel, which naturally introduces extra computation from simultaneous draft expansion and verification. To quantify this cost, we vary the number of explored blocks and generation lengths (Fig. 5a,b). Figure 5 reports results on HumanEval using LLaDA-7B (Instruct). The "base" corresponds to the Fast-dLLM (Dual Cache) baseline. On A800 GPUs, BlockSpec achieves clear latency reductions by leveraging tree-based parallelism, while on H800 the gap narrows because higher compute make the baseline itself more efficient, though BlockSpec still provides higher throughput.

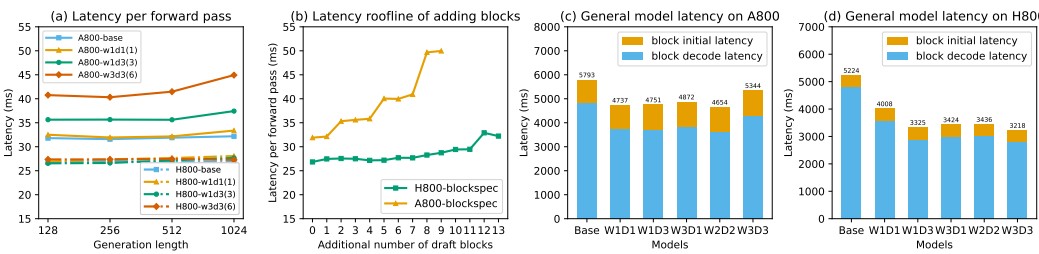

Figure 5: Analysis of computing overhead and latency. Base refer to Fast-dLLM (Dual Cache).

The key trade-off is that exploring more draft blocks increases parallel decoding but also adds per-step verification overhead. Since BlockSpec typically reduces the total number of decoding steps by 30–40%, additional blocks remain beneficial as long as the per-step latency increase stays within this range. Our roofline-style analysis in As shown in Fig. 5(a,b)shows that, on A800, exploring four extra blocks keeps the overhead acceptable, whereas stronger hardware can sustain larger structures such as W3D3 without slowdown. Overall, the end-to-end latency reflects a balance between fewer decoding iterations and higher per-step cost. As shown in Fig. 5(c,d), W2D2 achieves the best trade-off under A800 settings, while larger configurations (e.g., W3D3) become advantageous when additional parallelism can be utilized effectively. We report A800-based results primarily for consistency with prior baselines.

## 5 LIMITATIONS

Our proposed Block Speculation accelerates parallel decoding in low-confidence tokens of dLLMs through tree-based multi-step lookahead, but it has two main limitations. First, the acceptance rate of draft trajectories remains insufficiently understood and requires stronger theoretical support. Second, the method introduces additional computation, and reducing this overhead is an important direction for future optimization. These limitations highlight the need for more principled draft generation and more efficient verification to further advance speculative decoding for dLLMs.

## 6 CONCLUSION

In this work, we introduce Block Speculation, the full speculative decoding framework for diffusion-based LLMs. Our approach builds on blockwise draft-tree generation and parallel verification, enabling effective decoding even under low-confidence conditions. Through this design, Block Speculation achieves both higher speculative acceptance rates and controlled computational complexity, establishing the feasibility of speculative decoding in dLLMs. Experiments across diverse reasoning and code-generation benchmarks show that our method not only reduces decoding steps but also delivers up to 1.3× gains in parallel efficiency over strong baselines. We hope this work serves as a foundation for future research on speculative decoding tailored to diffusion models, paving the way for more accurate and more efficient parallel inference methods for dLLMs.

## THE USE OF LARGE LANGUAGE MODELS(LLMS)

In this work, large language models (LLMs) were only used for auxiliary purposes, including paper writing refinement and assistance with code completion. All research contributions, including the core algorithm design, experimental implementation and the general writing of this paper, were conducted independently by the authors. More details are shown in AppendixA.5.

## REPRODUCIBILITY STATEMENT

To ensure the reproducibility of our work, we provide: (i) a complete description of the experimental setup, including all baselines with their source code and the exact software/hardware environment, as detailed in AppendixA.1; (ii) pseudocode covering both draft generation in Algorithm1 and parallel verification in AppendixA.6 (iii) small-scale experiments across multiple hardware platforms to validate robustness, reported in Table3;

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

# A APPENDIX

## A.1 DETAILED EXPERIMENTS CONFIGURATIONS

Our experiments are conducted on two representative diffusion-based large language models: LLaDA-Instruct-7B and Dream-Instruct-7B. As baseline parallel decoding frameworks, we adopt Fast-dLLM and its dual-cache variant. To evaluate our method, we focus on mathematical reasoning and code generation tasks, which are widely used for generative model benchmarking. Specifically, we use GSM8K with MATH for math problem solving and HumanEval with MBPP for code generation, following the standard evaluation settings of the base models. All our models employ xFormers attention to support non-causal attention masks, while the baseline models use standard FlashAttention.

We note that existing dLLM baselines exhibit varying accuracy across different GPU architectures and CUDA versions. To ensure reproducibility and fairness, unless otherwise specified, all experiments are conducted on a single node with 8× NVIDIA A800 GPUs, using CUDA 12.6.3. This observation highlights the sensitivity of diffusion-based models to numerical precision. Therefore, for consistency, we re-conducted all experiments using the official open-source code released by the respective models. In most experiments, we fix the generation length to 512 tokens with block size = 32. In addition, under multi-GPU settings, we report averaged counts among gpus to obtain consistent single-card inference performance.

Fast-dLLM(dual-cache) serve as our primary baselines. BlockSpec adopts the same dual-cache strategy. At the beginning of each block's speculative decoding, we compute the full KV states for

both the prefix (verified tokens) and the suffix (remaining [MASK] tokens). These KV states are then frozen for the entire block and reused across all draft expansions and verification passes. During decoding, only the KV entries corresponding to the actively explored draft tokens are updated, while all other KV entries remain unchanged.

## A.2 TREE NODE EXPANSION STRATEGIES

In this section, we provide a detailed account of the design choices underlying our tree node expansion strategy for constructing draft trajectories. Each node corresponds to a draft partial decoding trajectory, and its children represent candidate token extensions. The expansion behavior is controlled by three key dimensions: 1) Branching factor (top-$k$ vs. top-1): the number of children instantiated when expanding a node; 2) Growth pattern (full-tree vs. left-biased): whether expansion proceeds symmetrically or favors left (higher-confidence) continuations; 3) Duplication policy (duplicate-allowed vs. duplicate-free): whether sibling nodes may share the same candidate token.

These dimensions jointly shape the trade-off between trajectory coverage and the computational overhead introduced by verifying additional draft blocks. Larger branching factors or more symmetric growth typically improve the likelihood of covering high-confidence continuations, but at the cost of increased verification workload.

Through extensive empirical evaluation, we adopt a simple yet highly effective hybrid strategy: (1) the root node expands using top-$k$ branching to preserve multiple plausible candidate trajectories, and (2) all deeper layers expand via top-1 branching, yielding a left-biased tree without duplicated siblings. This design induces a confidence-prefer expansion in deeper layers while maintaining exploratory breadth at the root.

As shown in the ablation study (Table 4), expanding more nodes generally increases the acceptance rate but does not substantially reduce the total number of decoding steps; instead, the additional nodes introduce significant per-step latency due to the extra verification overhead. We also note that in the duplicated-node variant, our implementation fully shares the draft computation—its actual compute cost remains equivalent to a six-node tree rather than nine—which explains why its empirical performance is close to the hybrid strategy; the only extra cost arises during verification. Based on these observations, the hybrid strategy provides the most favorable balance between acceptance rate and per-step computation, resulting in the strongest overall inference performance among all evaluated configurations.

| Expansion Strategy | Tree Config(num_block) | Step-1 AR | Step-2 AR | Step-3 AR | Avg. Steps / Answer | Avg. Latency / Step(s) | Avg. Overall Latency(s) |
|---|---|---|---|---|---|---|---|
| Hybrid | W2D2 (3) | 0.78 | 0.56 | – | 124.0 | 0.043 | **5.3** |
| Hybrid + top2 expansion | W2D2 (4) | 0.80 | 0.65 | – | 123.9 | 0.045 | 5.6 |
| Hybrid + full tree | W2D2 (4) | 0.80 | 0.61 | – | 124.9 | 0.045 | 5.6 |
| Hybrid + duplicated | W2D2 (4) | 0.80 | 0.61 | – | 125.0 | 0.043 | 5.3 |
| Hybrid | W3D3 (6) | 0.83 | 0.59 | 0.43 | 120.1 | 0.046 | **5.5** |
| Hybrid + top2 expansion | W4D3 (8) | 0.84 | 0.69 | 0.31 | 119.3 | 0.054 | 6.4 |
| Hybrid + full tree | W3D3 (9) | 0.83 | 0.60 | 0.38 | 120.3 | 0.055 | 6.6 |
| Hybrid + duplicated | W3D3 (9) | 0.83 | 0.62 | 0.37 | 119.8 | 0.047 | 5.6 |

Table 4: Ablation of node expansion strategies on HumanEval with gen_length = 512 on A800. "Step-1 AR" refers to accept rate in step 1 (level 2). The hybrid strategy provides the best trade-off between trajectory coverage and computational efficiency.

A visual comparison of the three dimensions—branching factor, growth pattern, and duplication policy—together with the resulting hybrid expansion structure is shown in Figure 6.

Overall, our hybrid design—*top-k at root, top-1 in deeper layers, and no duplicate nodes*—achieves optimal acceptance while maintaining superior decoding efficiency, making it the most practical choice among the strategies we evaluated.

## A.3 INTER-BLOCK SPECULATION

Inter-block speculation extends speculative decoding from a single block to a pair of *consecutive* blocks. The motivation arises from an empirical observation: once the current block has been partially decoded and sufficient contextual information has stabilized, the early tokens of the next block often already achieve high confidence due to the inherently parallel denoising behavior of diffu-

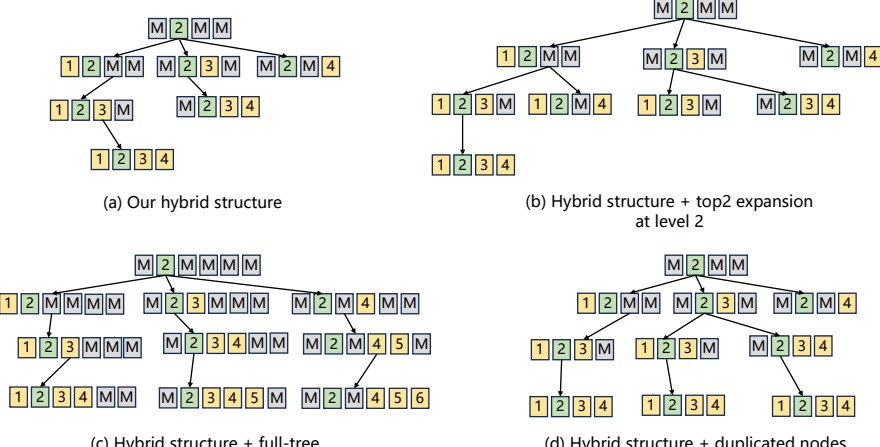

Figure 6: Visualization of different tree expansion strategies. (a) illustrates our final hybrid design, which forms a left-biased tree without duplicated sibling nodes. (b) shows a variant where each layer expands at least the top-2 candidates at level 2. (c) depicts a full-tree expansion, leading to a symmetric and much larger structure. (d) demonstrates a duplicated-node variant, where sibling nodes may contain identical target draft nodes; in implementation, we reuse the same draft computation for duplicated nodes, but during verification, each path is still validated independently.

sion models. Importantly, our method only performs speculation between the current block and its immediate successor; it does not attempt multi-block speculation across longer ranges.

Based on this observation, our inter-block speculation procedure consists of the following steps:

1. While decoding the current block with speculative drafting, we simultaneously run a forward pass on the next block and continuously monitor the confidence of its tokens.

2. Inter-block speculation is triggered when any token in the next block satisfies the trigger condition—either exceeding a predefined threshold $\tau$ or surpassing the confidence of unresolved positions in the current block.

3. Once triggered, we construct a compact draft tree for the next block (e.g., width 1, depth 3) following the same node expansion strategy used in intra-block speculation.

4. The draft trees of both the current and next blocks are then jointly verified under the interblock attention mask in a single parallel forward pass. Tokens in the next block are accepted only if their confidence exceeds $\tau$, after which both blocks continue decoding in parallel.

This mechanism exploits the early confidence rise in subsequent blocks and enables additional parallelism beyond intra-block speculation, while keeping overhead bounded and maintaining correctness. Figure 7 illustrates an example of inter-block speculation. While decoding the current block, the model concurrently monitors the token confidences of the next block; once these confidences exceed the trigger threshold, speculative decoding proceeds jointly across both blocks. It is worth noting that, due to the bidirectional attention mechanism of diffusion models, all suffix-masked blocks are fully preserved during speculative decoding.

**Trigger Condition**  During decoding of the current block $B_t$, we monitor

- $c_t^{(1)}$: the top-1 confidence score within block $B_t$,
- $c_{t+1}^{(1)}$: the top-1 confidence score within the next block $B_{t+1}$.

Inter-block speculation is triggered whenever

$$c_{t+1}^{(1)} > c_t^{(1)} \quad \text{or} \quad c_{t+1}^{(1)} > \tau, \tag{3}$$

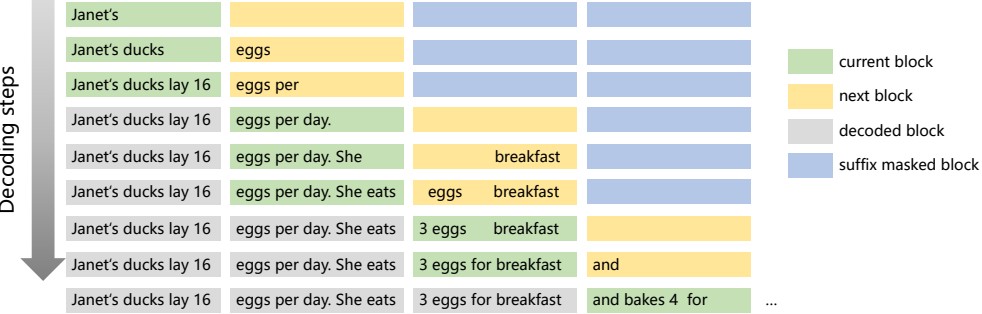

Figure 7: Visualization of inter-block speculative decoding. The figure illustrates decoding over four consecutive blocks. While decoding the current block, the algorithm simultaneously monitors the confidences of tokens in the next block. When high-confidence candidates (e.g., "eggs" in the second row or "breakfast" in the fifth row) appear, speculative decoding is activated jointly across both the current and next blocks.

where $\tau$ is a confidence threshold shared with intra-block speculation. This rule reflects the empirical observation that early positions in the next block often become reliable before difficult positions at the end of the current block.

**Draft-Tree Construction for the Next Block** For the next block $B_{t+1}$, we reuse the same tree-based draft generation strategy as intra-block speculation, with the following adaptation, as shown in Figure 8:

- The tree for $B_{t+1}$ is constructed independently, but always conditioned (blockwise causal attention) on the accepted root node of $B_t$ via the blockwise causal attention mask.
- To control overhead, we typically use a compact next-block tree (e.g., limiting width $W = 1$), which is empirically sufficient to harvest the high-confidence positions at the beginning of $B_{t+1}$.

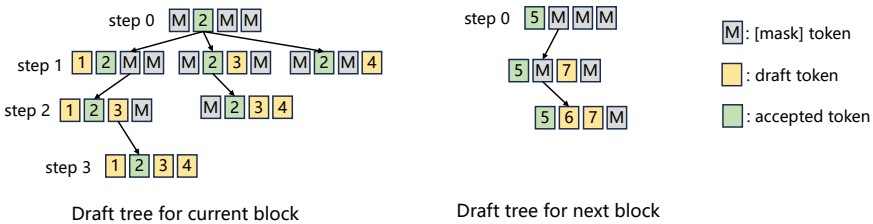

Figure 8: A sample of inter-block tree for current and next block.

**Inter-Block Attention Mask** The inter-block attention mask extends the blockwise formulation while preserving the bidirectional attention of inter-block roots. In our implementation, we do not impose a causal dependency between consecutive blocks. Instead, we retain full bidirectional visibility between the roots of the two adjacent blocks: all draft tokens of the next block are visible to the root of the current block, and all draft tokens of the current block are likewise visible to the root of the next block. In other words, the two blocks share each other's root Key/Value states, which

is consistent with prior findings in Fast-dLLM that the KV activations of prefix and suffix tokens remain highly similar across adjacent denoising steps. Meanwhile, drafts of the same block remain mutually invisible, ensuring that intra-block exploration is fully isolated.

**KV Cache** The KV states of prefix and suffix tokens (other than the current and next block) are refreshed at block boundaries under the dual-cache mechanism, remaining visible to both blocks throughout the decoding process, and thus maintaining the bidirectional denoising required by diffusion models.

A complete example of the resulting attention pattern is shown in Figure 9, illustrating the shared root visibility, the isolation of draft branches, and the blockly-cached prefix/suffix tokens.

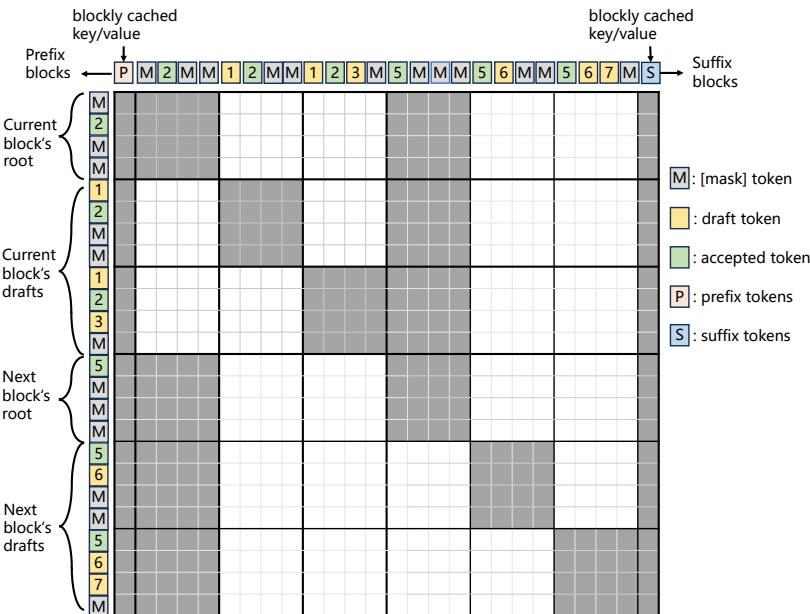

Figure 9: Visualization of inter-block mask attentions. The prefix and suffix tokens' KV states are refreshed at the beginning of each block via a full-length full-attention forward pass and then cached for blockwise reuse. In the inter-block setting, the suffix tokens do not include the tokens of the next block.

**Ablation study on interblock speculaton** We conduct an extensive ablation study to evaluate the potential and effectiveness of inter-block speculation for parallel decoding. Building on our default W2D2(3) draft-tree configuration, we progressively enable inter-block speculation with different amounts of additional lookahead blocks. As summarized in Table 5, introducing inter-block speculation consistently improves parallel decoding efficiency on top of the W2D2(3) baseline. Although each additional inter-block lookahead inevitably increases the per-step computational cost, we find that carefully controlling the number of extra speculative blocks yields a favorable balance between parallelism and overhead, leading to further end-to-end latency reductions.

| Inter-Block Setting | Tokens/Step ↑ | Avg. Steps / Answer ↓ | Avg. Latency / Step (s)↓ | Avg. Latency (s)↓ |
|---|---|---|---|---|
| W2D2(3), no inter-block | 3.83 | 124.0 | **0.043** | 5.3 |
| + Inter-block (1 draft block) | 4.29 | 109.1 | 0.046 | 5.0 |
| + Inter-block (2 draft block) | 4.38 | 106.7 | 0.047 | **5.0** |
| + Inter-block (3 draft block) | **4.53** | **103.0** | 0.049 | 5.1 |

Table 5: Ablation on inter-block speculation under the W2D2(3) draft structur for humaneval dataset on A800 GPUs. Increasing inter-block lookahead improves per-step parallelism and reduces the number of decoding steps, but alsointroduces additional cross-block computation and verification overhead. A moderate level of inter-block speculation

For a fair comparison, all measurements in this ablation are obtained on A800 GPUs. We expect the benefits of inter-block speculation to become even more pronounced on higher-end hardware such as H800, where the greater compute headroom can accommodate richer speculative structures and deeper inter-block exploration, which leave as an possible direction for future work.

**Pseudocode of Inter-Block Speculation** We summarize the inter-block mechanism in Algorithm 2.

---

**Algorithm 2:** Inter-Block Speculative Decoding

---

**Input:** Current block $B_t$ and next block $B_{t+1}$; token sequence $x$ with suffix `[MASK]` blocks;
diffusion LLM $p_\theta$; confidence threshold $\tau$; node-expansion configs $\text{Cfg}_{curr}, \text{Cfg}_{next}$
**Output:** Updated sequence $x$
**Function** InterBlockSpec$(x, B_t, B_{t+1})$**:**
   // Drafting for the current block
   $\mathcal{T}_t \leftarrow \texttt{BuildDraftTree}(x, B_t, \text{Cfg}_{curr}, \tau)$ ;
   // Probe next block while decoding $B_t$
   $(\ell_t, \ell_{t+1}) \leftarrow \texttt{ForwardWithMask}(x, \textit{InterBlockMask}(t, t+1))$ ;
   $c_t^{(1)} \leftarrow \texttt{Top1Conf}(\ell_t \textit{ on unresolved positions of } B_t)$ ;
   $c_{t+1}^{(1)} \leftarrow \texttt{Top1Conf}(\ell_{t+1} \textit{ on early positions of } B_{t+1})$ ;
   **if** $c_{t+1}^{(1)} > c_t^{(1)}$ **or** $c_{t+1}^{(1)} > \tau$ **then**
      // Build compact tree for $B_{t+1}$
      $\mathcal{T}_{t+1} \leftarrow \texttt{BuildDraftTree}(x, B_{t+1}, \text{Cfg}_{next}, \tau)$ ;
      // Joint verification across $B_t$ and $B_{t+1}$
      $\ell \leftarrow \texttt{ForwardWithMask}(x, \textit{InterBlockMask}(t, t+1))$ ;
      $x \leftarrow \texttt{JointVerify}(\mathcal{T}_t, \mathcal{T}_{t+1}, \ell, \tau)$ ;
   **else**
      // Fallback: intra-block only on $B_t$
      $\ell \leftarrow \texttt{ForwardWithMask}(x, \textit{IntraBlockMask}(t))$ ;
      $x \leftarrow \texttt{VerifySingleBlock}(\mathcal{T}_t, \ell, \tau)$ ;
   **return** $x$ ;

---

### A.4 COMPARISON WITH CONCURRENT WORKS

Concurrent to our submission, Agrawal et al. (2025) introduced Spiffy, another block-level speculative decoding framework. While both works explore blockwise speculation, our approach differs substantially in motivation, design, and empirical outcomes. Spiffy focuses on lossless verification via graph-based calibration, whereas our method is driven by a systematic study of low-confidence degeneration in diffusion LLMs, leading to a simpler tree-based draft structure centered on topk-confidence tokens under threshold. Our greedy blockwise verification, optional longest-path strategy, and support for inter-block speculation further improve efficiency while maintaining accuracy. In contrast, Spiffy operates only at the single-block level and requires full-graph traversal. We also evaluate against stronger baselines, including Fast-dLLM and its dual-cache variant, and consistently achieve markedly higher speedups (6.9×–13.6× vs. Spiffy's 5.18×). Overall, although developed independently, our framework provides a more targeted, efficient, and scalable solution to speculative decoding in diffusion LLMs.

In summary, while both works were developed independently and almost simultaneously, our method goes beyond Spiffy by targeting the low-confidence degeneration problem, designing more efficient draft and verification strategies, and extending to inter-block speculation. These differences translate into significantly stronger empirical results and a clearer path forward for scaling speculative decoding in diffusion LLMs.

### A.5 THE USE OF LARGE LANGUAGE MODELS (LLMS)

In this work, we used large language models (LLMs) for limited purposes. Specifically, we employed GPT-5 to assist with improving the paper writing, and we used the LLM agent IDE with

its integrated code models (including GPT- and Claude-based code completion) to programming syntax completion. All of core algorithmic designs, experimental implementations, and key code components presented in this paper were completed independently by the authors without reliance on LLMs.

### A.6   DETAILED PSEUDO CODE OF VERIFICATION MODULE

We design two parallel verification strategies: Greedy-Search and Longest-Search. In the Greedy-Search strategy, verification proceeds layer by layer, from left to right and select the first acceptable child immediately. In contrast, the Longest-Search strategy explores all candidate nodes at each layer and returns the longest valid sequence found in the draft trajectory tree. In practice, we observe that both strategies achieve comparable performance in terms of accuracy, and we finally use the Greedy-Search provides due to its reduced computational overhead.

---

**Algorithm 3:** Blockwise Parallel Verification (Greedy-Search)

---

**Input:** Draft trajectory tree $\mathcal{T}$ with child nodes $\mathcal{C}(n)$ for node $n$; draft candidates $\{d_i\}$; reference sequence $x$; initial decoded tokens $x_0$

**Output:** Accepted trajectory $\hat{x}_0$

Initialize current node $n \leftarrow 0$, result state $(x_0, mask)$ ;

**while** $\mathcal{C}(n)$ *is not empty* **do**

    $accepted \leftarrow$ False ;

    **foreach** *child* $c \in \mathcal{C}(n)$ **do**

        Obtain draft tokens $d_c$ for block $c$ ;

        Construct block mask $M_c$ to isolate positions of $d_c$ ;

        Compute verification conditions:

            (1) consistency with transfer indices of $x_0$

            (2) token agreement with current $x_0$ ;

        **if** *both conditions satisfied* **then**

            Update accepted tokens: $\hat{x}_0 \leftarrow \mathrm{merge}(x_0, d_c, M_c)$ ;

            Update transfer indices and masks ;

            $n \leftarrow c$, $accepted \leftarrow$ True ;

            **break**

    **if** *not accepted* **then**

        **stop verification** ;

**return** $\hat{x}_0$

---

### A.7   DETAILED PSEUDO CODE OF THRESHOLD DRAFT GENERATION

We further design and experiment with a threshold-based draft tree generation algorithm. Unlike the pure top-$k$ strategy, this approach augments deeper tree layers by not only selecting the top-$k$ candidates but also including all tokens whose confidence exceeds a predefined, layer-specific threshold $\tau_\ell$. The motivation is that incorporating multiple high-confidence candidates at each expansion step could improve recalls of candidate tokens. However, in our experiments, this strategy did not yield significant performance advantages over the simpler top-$k$ approach. We present it here as an alternative design to inspire future exploration.

---

**Algorithm 4:** Blockwise Parallel Verification (Longest-Search)

---

**Input:** Draft trajectory tree $\mathcal{T}$ with child nodes $\mathcal{C}(n)$ for node $n$; draft candidates $\{d_i\}$; reference sequence $x$; initial decoded tokens $x_0$
**Output:** Longest accepted trajectory $\hat{x}_0$ and path $\mathcal{P}$
**Function** DFS $(n, x_0, mask, \mathcal{P})$ **:**
    **if** $\mathcal{C}(n)$ *is empty* **then**
        **return** $(x_0, \mathcal{P})$ // Reached a leaf node
    Initialize $best\_path \leftarrow \mathcal{P}$, $best\_state \leftarrow (x_0, mask)$ ;
    **foreach** *child* $c \in \mathcal{C}(n)$ **do**
        Obtain draft tokens $d_c$ for block $c$ ;
        Construct block mask $M_c$ for $d_c$ ;
        Compute verification conditions:
            (1) consistency with transfer indices of $x_0$
            (2) token agreement with $x_0$ ;
        **if** *both conditions satisfied* **then**
            $\hat{x}_0^c \leftarrow \text{merge}(x_0, d_c, M_c)$ ;
            $(x_0', \mathcal{P}') \leftarrow$ DFS $(c, \hat{x}_0^c, mask, \mathcal{P} \cup \{c\})$ ;
            **if** $|\mathcal{P}'| > |best\_path|$ **then**
                $best\_path \leftarrow \mathcal{P}'$, $best\_state \leftarrow (x_0', mask)$ ;
    **return** $(best\_state, best\_path)$
$(\hat{x}_0, \mathcal{P}) \leftarrow$ DFS $(0, x_0, mask, [0])$ ;
**return** $\hat{x}_0, \mathcal{P}$

---

**Algorithm 5:** Tree-based Draft Trajectory Generation with Layer-wise Thresholds

---

**Input:** Previous model logits $\{\ell_t\}$, parameters $k, D, W$, thresholds $\{\tau_\ell\}_{\ell=1}^D$
**Output:** Draft trajectory tree $\mathcal{T}$
Initialize $\mathcal{T}$ with root node $T_0$ and empty token set $\text{tokens}(T_0)$ ;
**foreach** *token* $x$ **do**
    $c(x) \leftarrow \max(\text{softmax}(\ell_t(x)))$ ;
Candidate pool $C \leftarrow \{x\}$ ;
**for** $\ell = 1$ **to** $D$ **do**
    **if** $\ell = 1$ **then**
        $S \leftarrow$ top-$k$ tokens from $C$ ranked by $c(x)$ ;
        **foreach** $x \in S$ **do**
            create child node $n$ of $T_0$ with $\text{tokens}(n) \leftarrow \{x\}$ ;
    **else**
        **foreach** *node* $n$ *at layer* $\ell - 1$ **do**
            $S_{\text{topk}} \leftarrow$ top-$k$ tokens from $(C \setminus \text{tokens}(n))$ ranked by $c(x)$ ;
            **foreach** $x \in S_{topk}$ **do**
                create child node $m$ of $n$ with $\text{tokens}(m) \leftarrow \text{tokens}(n) \cup \{x\}$ ;
            $S_\tau \leftarrow \{x \in C \setminus \text{tokens}(n) \mid c(x) > \tau_\ell\}$ ;
            **if** $S_\tau \neq \emptyset$ **then**
                create child node $m_\tau$ of $n$ with $\text{tokens}(m_\tau) \leftarrow \text{tokens}(n) \cup S_\tau$ ;
    prune nodes of layer $\ell$ to width $W$ ;
**return** $\mathcal{T}$

---

## A.8 DECODING TRAJECTORY DEMOS

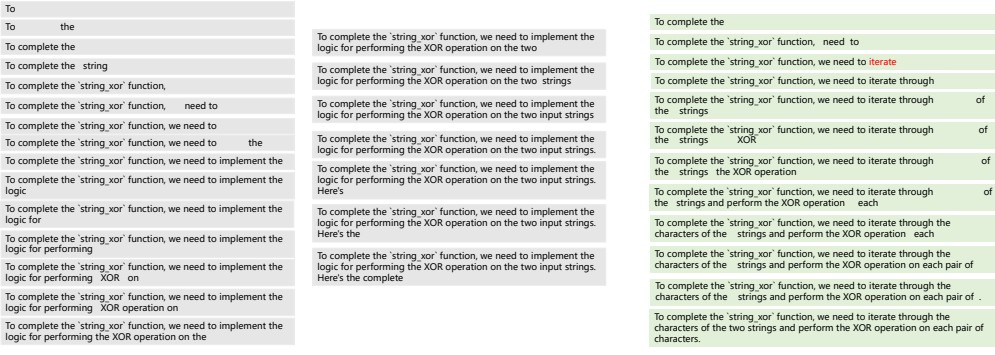

(a) Fast-dllm (dual cache) , 22 steps          (b) Our BlockSpec, 12 steps

Figure 10: Example Humaneval decoding trajectories of BlockSpec and Fast-dLLM (dual cache). For the first block, Fast-dLLM requires 22 denoising steps to finish decoding 32 tokens, whereas BlockSpec completes the same block in only 12 steps, showing substantially higher early-stage parallelism. Because the decoding trajectories differ, the two methods diverge at the red-highlighted position in the figure; in this case, BlockSpec's more aggressive trajectory yields the correct final answer, while Fast-dLLM ultimately produces an wrong output.

## A.9 GENERATION LENGTH ABLATION

To more systematically evaluate the behavior of BlockSpec under different generation lengths, we conduct an additional ablation study varying the number of generated tokens beyond the 512-token setting used in the main paper. Since our base model LLaDA is trained with a maximum context length of 4096 tokens (including bothprompt and response), we experiment with generation lengths of 128, 256, 512, 1024, and 2048 tokens. Table A.9 reports the end-to-end generation latency for both Fast-dLLM (Dual Cache) and our BlockSpec(W2D2) on Humaneval Dataset.

| Generation Length | Fast-dLLM (Dual Cache) | BlockSpec |
|---|---|---|
| 128 | 1.5 | 1.0 |
| 256 | 3.2 | 2.3 |
| 512 | 6.6 | 5.0 |
| 1024 | 10.2 | 8.7 |
| 2048 | 22.4 | 21.2 |

Table 6: End-to-end generation latency (seconds) under different output lengths.

We observe that BlockSpec consistently improves throughput across all sequence lengths. However, the relative gains become smaller as the generation length increases. This effect can be explained by the behavior of current discrete diffusion masked models: after a certain number of blocks, the model typically emits an [EOS] token, and all remaining positions in the sequence are filled with placeholder [EOS] tokens. Once [EOS] is generated, decoding each subsequent block requires only a single denoising step, since the entire block is deterministically populated with [EOS]. Because BlockSpec does not alter the model's intrinsic denoising process, both BlockSpec and the baseline collapse to this one-step-per-block pattern in the tail portion of long sequences, which naturally reduces the relative speed advantage at longer lengths.

To further illustrate this behavior, Figure 11 visualizes the average number of decoding steps across blocks (block size = 32). As shown in the figure, BlockSpec requires fewer steps in the early blocks where meaningful tokens are generated, while the later blocks converge to a single-step pattern due to early [EOS] emission. This explains why BlockSpec provides the strongest acceleration in

the informative part of the sequence, with diminishing gains once the decoding transitions into the `[EOS]`-only region.

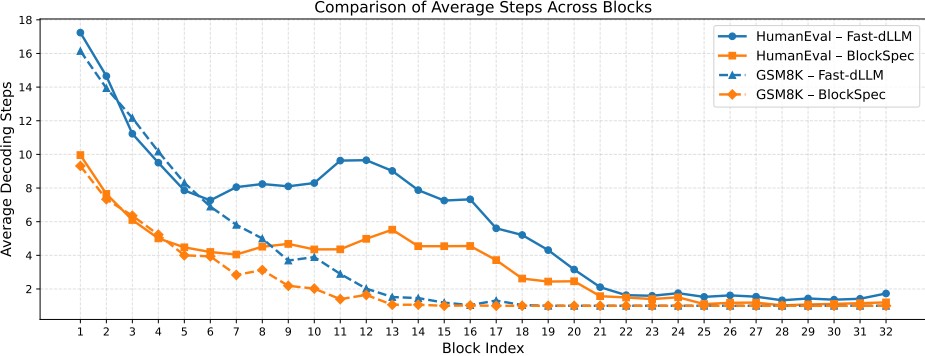

Figure 11: Visualization of the average number of decoding steps across blocks for models with length of 1024 on Humaneval and GSM8K. BlockSpec reduces decoding steps in most valid blocks.

### A.10 EXACTNESS AND FAILURE ANALYSIS

Classical autoregressive speculative decoding is mathematically exact: the speculative procedure always reproduces exactly the same tokens and decoding trajectory as the target model. In contrast, BlockSpec's block-wise speculative decoding for diffusion LLMs does not guarantee such exactness. The discrepancy primarily arises from the altered decoding trajectories introduced by block-level parallelism and speculative exploration.

To clarify this distinction, in this secton, we first isolate an idealized non-parallel setting and show that, under this restricted regime, BlockSpec is mathematically exact and provably aligned with the target model. This establishes a conceptual baseline illustrating when and why the procedure should succeed. We then explain how the introduction of parallel decoding and speculative branching alters the denoising trajectory and breaks this exactness in general.

Building on this theoretical foundation, the subsequent subsection evaluates failure cases empirically. Through controlled experiments on math-reasoning problems of increasing difficulty, we observe that BlockSpec may accumulate subtle trajectory deviations on harder instances, occasionally leading to small but measurable accuracy drops. These deviations remain limited in our evaluation benchmarks and can be mitigated by reducing the draft exploration depth. Nevertheless, they highlight an intriguing direction for future work toward developing more robust and potentially lossless speculative parallel decoding methods for diffusion LLMs.

### A.10.1 EXACTNESS ANALYSIS

Classical autoregressive speculative decoding guarantees mathematical exactness, but this guarantee does not hold for our block-wise speculative decoding in diffusion LLMs. In this subsection, we isolate an idealized setting: single-block, 1 token per step, and prove that under these conditions BlockSpec is mathematically exact. This provides a useful reference point for understanding where failures and inconsistencies arise.

**Proposition.** In a single-block setting, with cached prefix–suffix context and no inter-block speculation, if both decoders select the top-1 confidence token at every step, then BlockSpec produces a decoding trajectory that is mathematically identical to the trajectory of the target model (Fast-dLLM with dual cache).

**Notation.** Let $p$ be the target model which refers the Fast-dLLM(Dual Cache) here, and $C$ is the cached prefix–suffix context (all tokens outside the current block). The block size is $B$.

In each block, the target model produces a sequence of $(position, token)$ pairs:

$$(i_n, y_n), \qquad n = 1, \ldots, B,$$

by selecting:

$$(i_n, y_n) = \arg\max_{i,t} p(x_i = t \mid C, \ (i_1, y_1), \ldots, (i_{n-1}, y_{n-1})).$$

which refers to the top-1 confidence decoding strategy.

We note the trajectory as:

$$T_n = \{(i_1, y_1), \ldots, (i_n, y_n)\}.$$

**BlockSpec decoder.** BlockSpec proceeds in speculative steps $k = 1, 2, \ldots, K$. At step $k$ it accepts $S_k$ tokens forming:

$$\{(\hat{i}_1^{(k)}, \hat{y}_1^{(k)}), \ldots, (\hat{i}_{S_k}^{(k)}, \hat{y}_{S_k}^{(k)})\}$$

all chosen sequentially by top-1 decoding under $p$ with the tokens accepted so far. Flattening all steps yields

$$\{\{(\hat{i}_1^{(1)}, \hat{y}_1^{(1)}), \ldots, (\hat{i}_{S_1}^{(1)}, \hat{y}_{S_1}^{(1)})\}, \ldots, \{(\hat{i}_1^{(K)}, \hat{y}_1^{(K)}), \ldots, (\hat{i}_{S_K}^{(k)}, \hat{y}_{S_K}^{(k)})\}\} = \{(\hat{i}_1, \hat{y}_1), \ldots, (\hat{i}_B, \hat{y}_B)\}$$

We note the trajectory of BlockSpec as:

$$\hat{T}_n = \{(\hat{i}_1, \hat{y}_1), \ldots, (\hat{i}_n, \hat{y}_n)\}.$$

Our goal is to prove that every position of $T_n$ and $\hat{T}_n$ matches:

$$(i_n, y_n) = (\hat{i}_n, \hat{y}_n), \qquad \forall n.$$

**Theorem 1 (Exactness of cached context and top-1 decoding BlockSpec)** *Under cached prefix–suffix context, no inter-block speculation, and deterministic top-1 decoding, BlockSpec and the target decoder produce identical trajectories:*

$$(i_n, y_n) = (\hat{i}_n, \hat{y}_n) \qquad \forall n = 1, \ldots, B.$$

**Proof.** We induct over the speculative step $k$ of BlockSpec.

Let $M_k$ denote the total number of accepted tokens after step $k$:

$$M_0 = 0, \qquad M_k = M_{k-1} + S_k.$$

Thus the BlockSpec prefix after step $k$ is $\widehat{T}_{M_k}$.

**Base case.** Before step $k = 1$,

$$\widehat{T}_{M_0} = \widehat{T}_0 = \emptyset = T_0.$$

**Induction hypothesis.** Assume

$$\widehat{T}_{M_{k-1}} = T_{M_{k-1}}.$$

Because (i) all outside-block tokens are cached in $C$, (ii) no inter-block speculation occurs, and (iii) BlockSpec never modifies $C$, the conditional models coincide:

$$p_{\text{spec}}(x_i = t \mid C, \widehat{T}_m) = p(x_i = t \mid C, \widehat{T}_m) \quad \forall m < B.$$

Thus both decoders evaluate the *same* distribution at every substep.

**Induction step.** At speculative step $k$, BlockSpec accepts a path

$$(\hat{i}_1^{(k)}, \hat{y}_1^{(k)}), \ldots, (\hat{i}_{S_k}^{(k)}, \hat{y}_{S_k}^{(k)}),$$

chosen sequentially as

$$(\hat{i}_j^{(k)}, \hat{y}_j^{(k)}) = \arg\max_{i,t} p(x_i = t \mid C, \widehat{T}_{M_{k-1}}, (\hat{i}_1^{(k)}, \hat{y}_1^{(k)}), \ldots, (\hat{i}_{j-1}^{(k)}, \hat{y}_{j-1}^{(k)})).$$

On the target side, the next $S_k$ tokens are

$$(i_{M_{k-1}+1}, y_{M_{k-1}+1}), \ldots, (i_{M_k}, y_{M_k}),$$

selected from the same conditional distributions, because the induction hypothesis ensures

$$T_{M_{k-1}} = \widehat{T}_{M_{k-1}}.$$

For substep $j = 1$, both decoders apply $\arg\max$ to the same distribution

$$p(x_i = t \mid C, \widehat{T}_{M_{k-1}}),$$

so

$$(\hat{i}_1^{(k)}, \hat{y}_1^{(k)}) = (i_{M_{k-1}+1}, y_{M_{k-1}+1}).$$

For $j > 1$, the contexts remain identical because both procedures append the same $j - 1$ accepted tokens. Therefore

$$(\hat{i}_j^{(k)}, \hat{y}_j^{(k)}) = (i_{M_{k-1}+j}, y_{M_{k-1}+j}).$$

Hence after step $k$,

$$\widehat{T}_{M_k} = T_{M_k}.$$

**Conclusion.** After the final speculative step we have $M_K = B$, and therefore:

$$(\hat{i}_n, \hat{y}_n) = (i_n, y_n) \qquad \forall n.$$

$\square$

**Experimental Verification.** To empirically validate the theoretical result above, we implemented the exact special-case configuration required by the theorem: (i) cached prefix–suffix KV states, (ii) deterministic top-1 decoding, (iii) single-block decoding, and (iv) no inter-block speculation. Under this controlled setup, both the target model (Fast-dLLM Dual Cache) and BlockSpec evaluate identical conditional distributions at every substep.

We constructed parallel inference pipelines for both decoders and evaluated them on subsets of HumanEval and GSM8K. For each sample, we compared both the final decoded outputs and all intermediate predictive logits. Across all evaluated instances, the two trajectories matched exactly: every decoded token, every position, and every logit vector were identical. This provides strong empirical confirmation of the mathematical exactness established in the theorem.

| Dataset | Token Match Rate | Logit Mean Error |
|---|---|---|
| HumanEval | 100% | 0.00 |
| GSM8K | 100% | 0.00 |

Table 7: Empirical verification of exactness under cached prefix–suffix KV, top-1 decoding, and no inter-block speculation. Both decoded outputs and logits match perfectly across datasets.

### A.10.2 FAILURE CASES

Building on the previous subsection, we established that under an idealized configuration—cached prefix–suffix KV states, top-1 decoding, single-block execution, and no inter-block speculation— BlockSpec is mathematically exact and produces trajectories identical to the target Fast-dLLM decoder. This mirrors the classical guarantees of speculative decoding in autoregressive models. Consequently, any discrepancy between BlockSpec and the baseline mainly stem from the introduction of parallel decoding, which alters the decoding trajectory and may cause small deviations to accumulate, especially on tasks requiring precise intermediate reasoning.

To investigate where such trajectory-induced struggles emerge, we analyze BlockSpec's performance on the MATH dataset. This dataset spans diverse mathematical reasoning problems and includes fine-grained difficulty annotations from level 1 to 5, making it well-suited for isolating challenging cases. Table A.10.2 reports average accuracy for both the baseline and BlockSpec across a difficulty levels.

| Model | Level 1 | Level 2 | Level 3 | Level 4 | Level 5 | Average/Level |
|---|---|---|---|---|---|---|
| Fast-dLLM(Dual Cache) | 61.2% | 51.4% | 38.6% | 30.4% | 10.4% | 38.4% |
| BlockSpec | 61.3% | 52.2% | 38.7% | 25.1% | 9.2% | 37.4% |

Table 8: Accuracy of the baseline Fast-dLLM(Dual Cache) and BlockSpec across MATH difficulty levels. BlockSpec matches or exceeds the baseline on levels 1–3, while most divergence arises at level 4.

From the table, the overall difference between BlockSpec and the baseline is small ($\sim 1\%$), and nearly all of this gap originates from level-4 problems. Levels 1–3 show identical or slightly stronger performance, while the gap widens on level 4 and becomes marginal again on level 5. This pattern suggests that the performance drop is not due to systematic model degradation, but rather to specific reasoning chains where a small early trajectory deviation significantly impacts later steps.

To better understand these cases, we visualize representative failures (Figure 12). Each example highlights: (i) the point at which BlockSpec's parallel trajectory diverges from the target model, and (ii) the decoding state at that divergence. We observe that failures typically arise when a minor divergence in an early speculative step propagates across subsequent reasoning stages, eventually producing an incorrect intermediate symbol or misaligned reasoning branch. These errors are concentrated in problems that require multi-step symbolic precision, where small deviations cannot be easily corrected by later steps.

Consistent with our exactness analysis, since BlockSpec and the target model are provably identical when no parallelism is used, we conclude that these failures mostly from trajectory drift induced by parallel speculative decoding. On tasks with strict token-level reasoning requirements, such trajectory drift is more likely to accumulate, leading to occasional errors. Nevertheless, the overall accuracy drop remains small ($\sim 1\%$), indicating that BlockSpec is robust in practice despite introducing significantly more parallelism. Further improving the trajectory stability of parallel diffusion decoding could be a direction for future work.

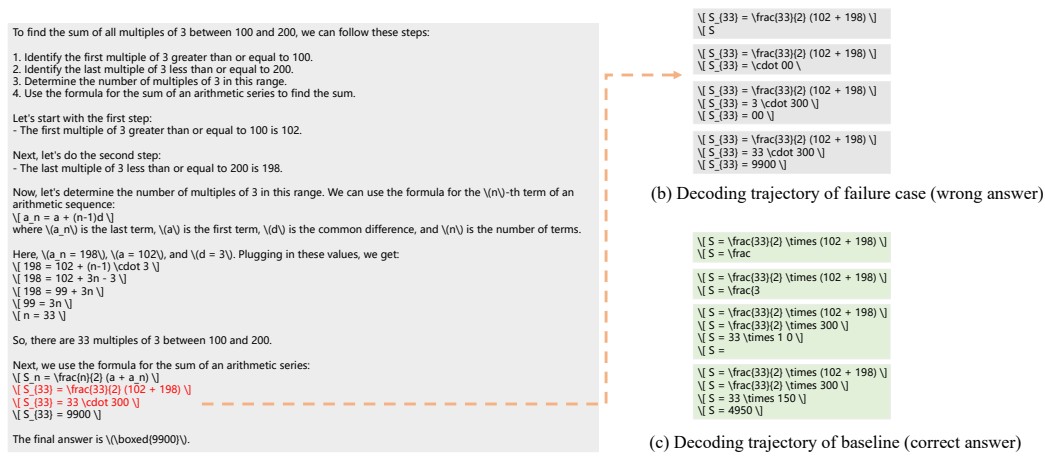

Figure 12: Failure case study of BlockSpec on mathematical reasoning. (a) A complete incorrect prediction generated by our method on a Level-4 problem from the Math dataset. (b) The corresponding decoding trajectory of BlockSpec, with the divergence point highlighted to illustrate where the reasoning chain begins to deviate. (c) The decoding path of the baseline Fast-dLLM (dual cache), which produces the correct final answer for the same input.

