# OpenReview forum: "BlockSpec: Blockwise Speculative Decoding for Diffusion LLMs"
_ICLR.cc/2026/Conference — Submitted to ICLR 2026_

### Official Review · Reviewer_DaQS · 2025-10-18

**Soundness:** 3
**Presentation:** 3
**Contribution:** 3
**Rating:** 6
**Confidence:** 4

**Summary:**

This paper introduces BlockSpec, a novel, training-free speculative decoding framework designed to mitigate the "low-confidence degradation" problem in diffusion-based Large Language Models (dLLMs), where parallel decoding efficiency collapses under uncertainty. BlockSpec addresses this by exploring multiple decoding trajectories simultaneously using a new tree-based generation strategy to create candidate token blocks in parallel. These blocks are then efficiently validated through a corresponding blockwise verification mechanism tailored for the any-order, bidirectional nature of dLLMs, with performance further enhanced by intra- and inter-block speculation. Experimental results demonstrate that this approach significantly reduces decoding steps, achieving up to a 14x speedup over vanilla dLLMs and a 1.3x improvement over strong baselines, establishing a more robust and efficient inference paradigm.

**Strengths:**

The proposed BlockSpec in the paper is a novel training-free decoding acceleration strategy that attempts to consider more branches to alleviate the degradation problem in parallel decoding. Experimental results show that this method is effective on different datasets, different models, and different hardware.

**Weaknesses:**

1. In Section 3.3, the authors state that “…allows key–value caching on the prompt…”. However, the rest of the paper (including the experimental settings) does not clarify whether KV caching was actually used. If KV cache was indeed applied, the authors should provide its design details and implementation description.

2. The stated motivation of the paper is to address the “low-confidence degradation problem of parallel decoding.” However, intuitively, the proposed method should lead to better performance, which is not reflected in the experimental results. I suggest that the authors revise the Introduction to better align it with the presented method and findings.

3. In Figure 3(3), for the current block, the subsequent masked tokens (the rightmost three blocks with ellipses) should be invisible. Is this a plotting error? This visualization seems to indicate that the model is aware that this is the final block to be generated, which could substantially influence the effective generation length and thereby degrade model performance.

**Questions:**

1-2. See weakness 1 and 3.
3. Although Section 3.4 introduces the concept of inter-block speculation, it still lacks sufficient implementation details. For example, it does not appear in any of the provided pseudocode segments. My question is: if both the current block and the next block are generated simultaneously, and each block is supposed to attend to the prompt as well as all preceding blocks, then which branch of the current block should the next block attend to?

---

> ### Author Response · Authors · 2025-11-19
>
> Q1: W1:In Section 3.3, the authors state that “…allows key–value caching on the prompt…”. However, the rest of the paper (including the experimental settings) does not clarify whether KV caching was actually used. If KV cache was indeed applied, the authors should provide its design details and implementation description.
>
> Response to Q1:
>
> We thank the reviewer for raising this important clarification. Yes—our method does use KV caching, and the caching mechanism we adopt is essentially the same dual-cache design introduced in Fast-dLLM (Dual Cache), which serves as our baseline. In the original submission, we did not provide sufficient detail on this component due to space constraints, and we fully agree that this omission could lead to confusion.
>
> In the revised manuscript, we have added a dedicated section (Sec. 3.4) that explicitly describes how suffix tokens and KV caching are handled in BlockSpec. To further highlight the relationship between our method and the dual-cache baseline, we also expanded the explanation of Fast-dLLM (Dual Cache) in Sec. 4.1 (highlighted in yellow).
>
> Briefly, our KV-cache mechanism operates as follows: At the start of each block’s speculative decoding, we recompute the full-attention KV states for 1) the prefix (prompt + previously decoded tokens), 2) the suffix (remaining [MASK] tokens), and 3) the current block root.
>
> The prefix and suffix KV states are then cached and kept fixed throughout all decoding steps within the block. These cached states are reused for all draft expansions and verification passes, and only the KV entries corresponding to the actively explored draft tokens are iteratively updated. This behavior matches the dual-cache assumption of Fast-dLLM that prefix/suffix KV activations remain highly similar across adjacent denoising steps.
>
> For full details—including diagrams and implementation notes—please refer to the newly added Sec. 3.4 in the revised version. We sincerely appreciate the opportunity to clarify our KV-cache design, and we hope the revised manuscript fully resolves any confusion regarding this component.
>
> Q2: W3: In Figure 3(3), for the current block, the subsequent masked tokens (the rightmost three blocks with ellipses) should be invisible. Is this a plotting error? This visualization seems to indicate that the model is aware that this is the final block to be generated, which could substantially influence the effective generation length and thereby degrade model performance.
>
> Response to Q2:
> We appreciate the reviewer for this accurate observation. Yes—this was indeed a plotting error. In our actual implementation, the model never hides or discards the suffix [MASK] tokens during blockwise speculative decoding. All remaining masked positions are preserved and included in the computation, exactly as in Fast-dLLM (Dual Cache). Their KV states are refreshed once at each block boundary and then reused throughout the block, contributing both to correctness and to per-step efficiency.
>
> In the original figure, we omitted the suffix tokens for visual simplicity when illustrating the draft blocks, which unintentionally suggested that the model knew it had reached the final block. We have corrected this in the revised manuscript as: 1) Figure 3(3) has been updated to properly include all suffix [MASK] tokens, 2) Sec. 3.4 now explicitly describes our suffix-handling and KV-cache mechanism, and 3) Appendix A.3 provides an updated inter-block attention-mask diagram where suffix visibility is also shown.
>
> We sincerely appreciate the reviewer for pointing out this issue and for giving us the opportunity to improve the correctness and clarity of our visualizations.

---

> ### Author Response · Authors · 2025-11-19
>
> Q3: Although Section 3.4 introduces the concept of inter-block speculation, it still lacks sufficient implementation details. For example, it does not appear in any of the provided pseudocode segments. My question is: if both the current block and the next block are generated simultaneously, and each block is supposed to attend to the prompt as well as all preceding blocks, then which branch of the current block should the next block attend to?
>
> Response to Q3:
> Appreciate for raising this question. The reviewer is absolutely right that the initial submission did not provide enough implementation detail for inter-block speculation, largely due to page-limit constraints. We have significantly expanded this part of the paper in the revised version.
>
> In the updated manuscript, Appendix A.3 now provides a complete exposition of inter-block speculation, including its motivation, triggering conditions, draft-tree structure, step-by-step decoding demo, and the full attention-mask design. We have also added clarifying text in the main paper (Sec. 3.5).
>
> Regarding the reviewer’s specific question—which branch of the current block does the next block attend to?—we apologize that the original text did not make this explicit. The answer is: The next block attends only to the root (i.e., the accepted version) of the current block, not to any draft branches.
>
> All draft variants within the current block remain mutually invisible to the next block. The next block receives a single, stable root representation from the current block. This root–root connection is implemented as full bidirectional visibility, as detailed in Appendix A.3.
>
> This design avoids ambiguity—there is only one representation from the current block that the next block can rely on—and preserves the stability required for accurate denoising. Prefix and suffix KV states remain blockly-cached and visible to all blocks, as described in Sec. 3.4.
>
> We sincerely thank the reviewer for highlighting this gap. We hope the expanded description and newly added diagrams in Appendix A.3 can better clarify how inter-block speculation is implemented.
>
> Q4: The stated motivation of the paper is to address the “low-confidence degradation problem of parallel decoding.” However, intuitively, the proposed method should lead to better performance, which is not reflected in the experimental results. I suggest that the authors revise the Introduction to better align it with the presented method and findings.
>
> Response to Q4:
> We thank the reviewer for this valuable comment. Our motivation is indeed to address the “low-confidence degradation problem of parallel decoding”, but we realize that our original phrasing may have unintentionally suggested that low-confidence tokens harm generation quality. What we aim to resolve is a different issue: Low-confidence tokens cause a sharp drop in decoding parallelism, not a drop in model accuracy.
>
> In existing diffusion-based parallel decoding methods, a low-confidence position typically forces the model to decode only one token per step, collapsing parallelism and significantly slowing down inference. BlockSpec is designed specifically to restore parallel decoding efficiency in these regions—from 1 token per forward pass to more than 2–4 tokens per forward pass depending on configuration—while keeping model accuracy.
>
> We appreciate the reviewer for this advice. And we think that our wording “degradation” may have introduced confusion. A term like “degeneration of parallelism” may be more accurate. To better align the introduction with our actual method and findings, we have revised the relevant motivation statements(highlighted in green), clarified that the issue concerns efficiency rather than accuracy, and polished the introduction accordingly.

---

> > ### Comment · Reviewer_DaQS · 2025-11-21
> >
> > I have carefully reviewed the revised manuscript. The paper correctly highlights an important limitation of parallel decoding: in many cases, the model still effectively generates only one token at a time. However, such uncertainty does not necessarily imply incorrectness, and recognizing this distinction is a valuable insight. That said, the paper still has the following limitations:
> >
> > 1. **Tree verification.** Apart from replacing node-level tokens with blocks, the proposed technique appears to be a rather direct adaptation of speculative decoding methods used in autoregressive models such as Eagle and Medusa. I recommend that the authors include a dedicated paragraph clarifying the conceptual and technical differences between their method and prior work.
> >
> > 2. **Block-size constraints.** Because the block size cannot be set too large, the feasible decoding region becomes restricted. The original motivation of dLLMs is to break the strict left-to-right generation order of autoregressive models and enable arbitrary-order generation. However, in BlockSpec, the block-size constraint forces generation to remain within the current or next block, and this size cannot be increased substantially (e.g., it is typically set to 32). As a result, the method still effectively enforces a left-to-right generation pattern, which undermines model performance. I suggest evaluating the approach on a global-planning task such as Sudoku to diagnose this issue.
> >
> > Considering both the strengths and the remaining limitations of this work, I am inclined to maintain my previous score.

---

> > > ### Author Response · Authors · 2025-11-26
> > >
> > > We sincerely thank the reviewer for these thoughtful comments and appreciate the opportunity to further clarify the two remaining limitations raised in the review.
> > >
> > > Limitation 1: Tree verification and the relation to AR speculative decoding
> > >
> > > We fully agree that speculative decoding in AR models (e.g., Eagle, Medusa) provides an important conceptual precedent, and clarifying the similarities and differences is valuable. While our work is indeed inspired by classical speculative decoding, its migration into diffusion LLMs is beyond a direct adaptation. dLLMs differ fundamentally from AR models due to bidirectional attention, multi-token parallel decoding, and blockwise generation. These properties require substantial redesign for speculative decoding to function at all.
> > >
> > > In our method, the blockwise draft tree is only one of several dLLM-specific components. In particular, our framework introduces other three key innovations:
> > >
> > > a) Blockwise speculative decoding framework that supports arbitrary-order token speculation. Our draft construction operates over trajectories rather than left-to-right token chains, enabling exploration and parallel verification over any token order and arbitrary numbers of unresolved positions—capabilities that AR speculative decoding cannot express.
> > >
> > > b) A principled analysis of low-confidence degeneration and a tailored draft trajectory generation strategy. We identify that low-confidence positions systematically collapse parallel decoding in dLLMs. Based on this observation, our draft-generation mechanism explicitly collects and accelerates these positions while preserving correctness via parallel verification—this motivation is unique to diffusion models and paralle decoding methods.
> > >
> > > c) Inter-block speculation—a new form of segmented parallel decoding unique to dLLMs. Inter-block speculation leverages dLLMs’ bidirectional visibility and blockwise denoising to enable parallel lookahead across blocks. This segmented multi-block parallelism has no analogue in AR speculative decoding and, to our knowledge, BlockSpec is the first work to introduce this strategy.
> > >
> > > For these reasons, BlockSpec is not a simple transfer of AR speculative methods. Instead, our design originates from analyzing low-confidence degeneration in dLLMs and developing a speculative decoding framework that is compatible with arbitrary-order denoising and blockwise attention. We conducted extensive empirical exploration to derive the current speculation strategies. As one of the earliest attempts at speculative decoding for diffusion LLMs, we hope our findings can inform future works like improved draft structures, more efficient block verification, and more precise candidate selection strategies.
> > >
> > > We appreciate the reviewer’s comment, and in the revision we will add a dedicated paragraph(as shown below) clearly contrasting BlockSpec with AR speculative decoding and explaining why dLLMs require a different design.
> > >
> > >
> > >
> > > Limitation 2. Block-size constraints and semi-autoregressive behavior
> > > The reviewer correctly notes that one motivation of dLLMs is to allow arbitrary-order generation. At the same time, prior work has consistently shown that inference-time semi-autoregressive scheduling—performing any-order denoising within blocks while moving left-to-right across blocks—is critical for stable performance.
> > >
> > > As demonstrated in Large Language Diffusion Models [1], the gap is dramatic:
> > >
> > > - LLaDA-Instruct, unrestricted any-order decoding: 12.9
> > >
> > > - LLaDA-Instruct, semi-autoregressive decoding: 73.8
> > >
> > > This substantial difference indicates that, despite being trained with arbitrary-order denoising, current dLLMs rely heavily on semi-autoregressive progression during inference. BlockSpec is fully aligned with this established decoding paradigm. Our method does not introduce new left-to-right constraints; rather, it operates within the same blockwise progression already used in existing dLLMs and strengthens parallelism inside that structure through speculative decoding.
> > >
> > > We appreciate the reviewer’s suggestion to include a global-planning task such as Sudoku. While integrating such tasks is beyond the scope of our current study—which focuses on understanding and improving inference behavior under practical dLLM decoding paradigms—we agree it is an interesting direction, and we plan to explore it in future work.

---

> > > > ### Author Response · Authors · 2025-11-26
> > > >
> > > > New paragraph to be added in the revision (final version):
> > > >
> > > > Relationship to AR speculative decoding. Speculative decoding in autoregressive (AR) models typically relies on fixed left-to-right generation and causal masking, allowing token-by-token draft construction and one-step parallel verification. However, these mechanisms do not directly transfer to diffusion LLMs, whose decoding is governed by bidirectional attention, multi-token denoising, and blockwise semi-autoregressive progression. BlockSpec introduces several design elements specifically tailored to this setting: (i) a blockwise speculative decoding framework capable of constructing and verifying arbitrary-order denoising trajectories; (ii) a draft-generation mechanism motivated by the low-confidence degeneration issue unique to dLLMs; and (iii) an inter-block speculative formulation that exploits bidirectional visibility across adjacent blocks to enable segmented multi-block lookahead—an ability absent in AR models. Together, these components constitute a speculative decoding framework fundamentally different from AR approaches, reflecting the architectural and behavioral differences of diffusion-based LLMs.

---

> ### Comment · Reviewer_DaQS · 2025-11-26
>
> Thank you for the new response and the revised manuscript. All of my previous concerns have been fully addressed. The current version is well written and presents its arguments clearly. At this point, the only remaining limitation seems to be that the number of tree nodes cannot be very large; for instance, according to Line 399, most cases contain only three nodes. The acceleration benefits of speculative decoding largely come from maximizing GPU utilization, and on less powerful GPUs, a small number of tree nodes may limit the achievable speedup. I would be willing to keep my positive evaluation.

---

### Official Review · Reviewer_XYVg · 2025-10-24

**Soundness:** 2
**Presentation:** 2
**Contribution:** 1
**Rating:** 4
**Confidence:** 4

**Summary:**

This paper proposes BlockSpec, a training-free, blockwise speculative decoding method for diffusion language models (dLLMs). Through tree-structured trajectory generation and block-level parallel verification, the method achieves efficient decoding with high parallelism even under low-confidence conditions. The paper further introduces intra-block and inter-block speculation mechanisms, evaluates performance on reasoning and code-generation benchmarks, and analyzes computational overhead and limitations, reporting up to a 14× speedup over a vanilla dLLM.

**Strengths:**

1. The proposed BlockSpec framework delivers substantial decoding acceleration, reducing iteration steps by up to 40% and achieving 7–14× speedups over vanilla diffusion LLMs, while maintaining comparable output quality on reasoning and code-generation benchmarks.

2. By combining tree-based trajectory generation with block-level parallel verification, the method effectively mitigates low-confidence degradation in diffusion LLMs, sustaining parallelism without compromising accuracy.

**Weaknesses:**

1. According to Fig. 3(3), all draft blocks are invisible to subsequent masked tokens. Thus, when unmasking a block, the process essentially reduces to dLLM-style generation with a generation length equal to the block size, which can lead to issues such as prematurely forcing an answer or emitting eos early. This suggests that the current blockwise attention mask is problematic.

2. Based on Tables 1 and 2, BlockSpec’s high throughput (TPS) speedup seems attributable not only to fewer forward passes but also to reduced matrix computation induced by the specialized attention mechanism. However, the experiments and appendix do not clearly isolate or quantify this contribution.

**Questions:**

1. See Weakness 1.

2. What is the structure of the blockwise attention mask in inter-block speculation.

3. The experiments appear to show that a blockwise attention mask that breaks the bidirectional attention mechanism can still yield generation that is nearly lossless relative to full bidirectional attention. Have the authors investigated the underlying reasons for this?

---

> ### Author Response · Authors · 2025-11-19
>
> Q1: According to Fig. 3(3), all draft blocks are invisible to subsequent masked tokens. Thus, when unmasking a block, the process essentially reduces to dLLM-style generation with a generation length equal to the block size, which can lead to issues such as prematurely forcing an answer or emitting eos early. This suggests that the current blockwise attention mask is problematic.
>
> Response to Q1:
> We appreciate the reviewer for raising this important point. The concern is caused by a misleading illustration in Fig. 3(3), rather than by the actual design of our method. In our implementation, we strictly follow the Fast-dLLM (Dual Cache) blockwise KV-cache mechanism.
>
> At the beginning of each block’s speculative decoding, we run a full-attention forward pass and compute complete KV states for (i) all prefix tokens (prompt + decoded tokens), (ii) all suffix [MASK] tokens, and (iii) the current block. These KV states for prefix and suffix tokens are then cached and remain fixed throughout the entire block, just like Fast-dLLM(Dual Cache). During subsequent draft-tree expansions and verification, each draft token is visible to all cached prefix and suffix tokens, while only the KV entries associated with the explored draft tokens are updated. Therefore, draft decoding does not reduce to isolated dLLM-style generation; the  figure in initial submission simply failed to depict this correctly.
>
> In the revised manuscript, we have redrawn Fig. 3(3) to clearly show that:
>
> - all draft tokens are visible to all cached prefix and suffix states, and
>
> - draft tokens from different draft nodes are mutually invisible to avoid cross-node interference.
>
> We also acknowledge that our earlier wording of “blockwise causal attention” may have contributed to the confusion. The term was not an accurate description of the mechanisms used in our method.
>
> - In intra-block speculation, the correct term is blockwise individual attention:
>  draft tokens are visible to all blockly-cached prefix and suffix tokens, but remain mutually invisible across draft branches.
>
> - In inter-block speculation, we do not impose a causal dependency between consecutive blocks.
>  Instead, the roots of two adjacent blocks retain full bidirectional visibility: all draft tokens of the next block are visible to the current block’s root, and all draft tokens of the current block are likewise visible to the next block’s root. This shared-root visibility is consistent with Fast-dLLM’s observation that the KV activations of prefix/suffix tokens remain highly similar across adjacent denoising steps.
>
> - The only true causal structure in our masking scheme lies between the draft tokens of the decoding blocks and the blockly-cached prefix/suffix tokens. Prefix and suffix states are treated as the stable context(alse cached) that each decoding block condition on, while all draft-level interactions (both intra- and inter-block) avoid introducing additional causal constraints.
>
> To prevent further misunderstanding, we have updated the terminology in Sec. 3.1 and Sec. 3.3 and clarified the handling of suffix tokens and blockly-cached KV states. We also provide revised attention-mask diagrams in Appendix A.3 that accurately illustrate the intra-block, inter-block, and prefix/suffix visibility patterns.
>
> Q2: What is the structure of the blockwise attention mask in inter-block speculation.
>
> Response to Q2:
> Thanks for this suggestion. In the revised version, we have added a complete description of our inter-block speculation mechanism in Appendix A.3, including updated diagrams and detailed explanations of how visibility and KV caching are handled across adjacent blocks. The corresponding Inter-Block Attention Mask is also illustrated in Appendix A.3, providing a clear depiction of the masking rules used during inter-block draft expansion and verification.
>
> A direct textual response is:  The inter-block attention mask extends the blockwise formulation while preserving the bidirectional attention for inter-block roots. In our implementation, we do not impose a causal dependency between consecutive blocks. Instead, we retain full bidirectional visibility between the roots of the two adjacent blocks: all draft tokens of the next block are visible to the root of the current block, and all draft tokens of the current block are likewise visible to the root of the next block. In other words, drafts of two blocks share each other’s root Key/Value states, which is consistent with prior findings in Fast-dLLM that the KV activations of prefix and suffix tokens remain highly similar across adjacent denoising steps. Meanwhile, draft tokens across different blocks remain mutually invisible, ensuring that intra-block exploration is fully isolated.

---

> ### Author Response · Authors · 2025-11-19
>
> Q3: The experiments appear to show that a blockwise attention mask that breaks the bidirectional attention mechanism can still yield generation that is nearly lossless relative to full bidirectional attention. Have the authors investigated the underlying reasons for this?
>
> Response to Q3:
> The reviewer's question is very perceptive and important. Indeed, in principle, a model with a clear train–test attention mismatch would be expected to suffer noticeable accuracy degradation. We have conducted some  experiments, we are happy to share our explanations and findings here for more discussion.
>
> As discussed in Q1 and Q2(also shown in AppendixA.3), our method actually preserves substantial bidirectional structure in both suffix handling and inter-block interactions. Our design does not replace full bidirectional attention with a fully blockwise-causal mask. Instead: 1） all draft tokens remain fully visible to blockly-cached prefix and suffix tokens, and 2）adjacent blocks retain full bidirectional visibility between their root states. These preserved components of the bidirectional structure are crucial for maintaining generation quality and explain why our accuracy closely matches the Fast-dLLM baseline.
>
> This design choice is directly motivated by experiments that just confirming the reviewer’s insight --breaking bidirectional attention aggressively leads to dramatic degradation. We have tested two “fully causalized” alternatives: 1） applying strict blockwise causal attention (no bidirectional reuse of prefix/suffix KV), and 2） imposing a fully causal link between consecutive blocks (next block only sees current block → unidirectionally). Both variants suffered >15% accuracy drops, confirming that strong deviations from the bidirectional structure of diffusion LLMs severely impair model quality. These tests reinforced the necessity of retaining selective bidirectional visibility.
>
> Our implementation is guided by Fast-dLLM’s empirical finding that prefix/suffix KV activations remain highly similar across adjacent denoising steps. Leveraging this property, our method preserves bidirectional context in two key places: a) Prefix/suffix tokens: At each block boundary, we perform a full-attention forward pass to obtain fresh KV states and treat them as blockly-cached KV, shared across all decoding steps within the block. b) Inter-block roots: We retain full bidirectional visibility between the roots of the two adjacent blocks, allowing both blocks’ draft tokens to condition on a shared, stable representation.
>
> Together, these designs maintain much of the functional behavior of full bidirectional attention while still enabling efficient speculative parallel decoding. We appreciate the reviewer for raising this question.  It aligns closely with our own diagnostic observations and motivates us to further elaborate these mechanisms in the revised manuscript.
>
> Q4: W2: Based on Tables 1 and 2, BlockSpec’s high throughput (TPS) speedup seems attributable not only to fewer forward passes but also to reduced matrix computation induced by the specialized attention mechanism. However, the experiments and appendix do not clearly isolate or quantify this contribution.
>
> Response to Q4:
> We appreciate the reviewer for this very accurate observation. Indeed, yes,  the throughput improvements over vanilla llada/dream model come from two sources: (1) fewer denoising steps enabled by our proposed blockwise speculative parallel decoding, and (2) faster per-step computation, actually inherited from the dual KV-cache mechanism introduced by Fast-dLLM (Dual Cache).
>
> The latter is an important optimization by Fast-dLLM(Dual Cache): by reusing blockly-cached prefix and suffix KV states within each block, the attention computation becomes significantly cheaper than in the original LLaDA model. Since BlockSpec is built directly on top of Fast-dLLM (Dual Cache), our per-step efficiency benefits are essentially the same as those already established in that baseline. Our work focuses on improving parallel decoding efficiency, not introducing additional optimizations beyond the dual-cache design.
>
> We acknowledge that our initial submission did not emphasize this baseline component clearly enough. In the revised version, we have: 1) added a detailed explanation of the dual KV-cache mechanism in Sec. 3.4, and 2) clearly highlighted Fast-dLLM (Dual Cache) as our baseline in Sec. 4.1 (yellow highlighted), to avoid further ambiguity.
>
> To isolate the contribution of the KV-cache optimization, we can refer to the speed difference between Fast-dLLM and Fast-dLLM (Dual Cache) in the experiments. Since BlockSpec builds on the latter, we did not repeat this comparison in our ablations.
>
> We thank the reviewer for pointing this out—it helped us improve the clarity of our presentation. We hope that the revised manuscript resolves the confusion regarding the role of dual KV caching and clarifies how BlockSpec builds on this mechanism.

---

> > ### Comment · Reviewer_XYVg · 2025-11-25
> >
> > Thank you for the detailed responses and clarifications. I appreciate the effort the authors have put into addressing the concerns. However, I still identify two major issues that remain after the rebuttal:
> > 1.	Incremental contribution beyond Fast-dLLM (Dual Cache).
> > BlockSpec’s core addition appears to be a tree-structured intra-block and inter-block speculative schedule, a design pattern that is conceptually borrowed from ARM-style speculative decoding. While such scheduling is reasonable, the methodological novelty is modest, and most of the technical foundation—including the blockwise KV caching, reduced attention cost, and bidirectional prefix/suffix reuse—comes directly from Fast-dLLM (Dual Cache). As a result, the overall contribution does not seem sufficiently substantial.
> > 2.	Most of the observed speedup appears to come from Dual Cache rather than BlockSpec itself.
> > Based on the results in Tables 1 and 2 and the clarification in the rebuttal, the large throughput gains relative to vanilla dLLMs are largely inherited from the Dual Cache mechanism. The additional improvement brought by BlockSpec over Fast-dLLM (Dual Cache) is relatively small. This suggests that the core acceleration originates from Dual Cache, while BlockSpec’s incremental benefit is limited.
> > Given these remaining concerns, I am inclined to maintain my previous score.

---

> > > ### Author Response · Authors · 2025-11-26
> > >
> > > We sincerely thank the reviewer for the thoughtful follow-up. These questions address the core of our contribution, and we appreciate the opportunity to clarify them more precisely.
> > >
> > > Issue 1: Incremental contribution beyond Fast-dLLM (Dual Cache)
> > >
> > > We agree that BlockSpec is built on Fast-dLLM (Dual Cache). The additional descriptions of prefix/suffix KV caching and blockwise attention in the revised version were added only to resolve earlier confusion and are not claimed as contributions. They simply clarify the baseline substrate on which our method operates.
> > >
> > > Beyond these inherited components, BlockSpec introduces a new speculative decoding framework tailored specifically to diffusion LLMs, including several elements absent in Fast-dLLM and ARM-style speculative decoding:
> > >
> > > (a) Blockwise speculative decoding for arbitrary-order denoising. AR speculative decoding assumes left-to-right causal generation, while dLLMs must construct denoising trajectories over multi-token masked sequences. BlockSpec provides a new formalism that jointly handles token identity, position, and ordering under bidirectional attention.
> > >
> > > (b) A principled analysis and remedy for low-confidence degeneration. We show that Fast-dLLM frequently collapses to single-token decoding at low-confidence positions. This motivates our draft-token collection policy that explicitly targets these positions to sustain parallelism.
> > >
> > > (c) A carefully explored tree-structured draft strategy. Our tree design—root top-k expansion, deeper top-1, left-biased growth, duplication control, and depth–width trade-offs—results from extensive empirical exploration (Appendix A.2). Although inspired by AR speculative patterns, adapting them to arbitrary-order trajectories and much higher verification cost in dLLMs required substantial redesign. We hope these findings offer guidance for future dLLM speculative frameworks.
> > >
> > > (d) Inter-block speculation (a new form of segmented parallelism). Enabled by diffusion models’ bidirectional visibility, this cross-block lookahead has no analogue in AR models. To our knowledge, BlockSpec is the first to introduce such a mechanism.
> > >
> > > Issue 2: Most speedup comes from Dual Cache rather than BlockSpec
> > >
> > > We agree that Fast-dLLM (Dual Cache) introduced the first major acceleration for dLLMs, and we do not claim it as our contribution. Our work builds upon this strong baseline—as do nearly all recent dLLM inference methods.
> > >
> > > Our method sits on top of this baseline. Within this setting, BlockSpec provides additional speedup on an already highly optimized system, which is substantially more challenging than accelerating vanilla dLLMs. Despite this, BlockSpec achieves: 1) ≈40% reduction in decoding steps, 2) ≈30% decrease in end-to-end latency, 3) tokens-per-step from 2.6 → 4.3 (+65% increase)
> > >
> > > This increased parallelism is meaningful, as tokens-per-step directly upper-bounds achievable speedup, and future optimized verification kernels can further amplify these benefits. We report speedups over vanilla models because they remain the standard comparison in diffusion LLM literature; e.g., DPad [1] achieves ~4.14–10.32×, while BlockSpec reaches 7–14× with similar accuracy.
> > >
> > > Importantly, Dual Cache and BlockSpec optimize orthogonal dimensions: 1）Dual Cache reduces redundant computation (systems-level). 2）BlockSpec reduces required denoising iterations (algorithmic-level). Their gains accumulate, and BlockSpec’s improvements remain visible even atop Fast-dLLM (Dual Cache).
> > >
> > > We appreciate the reviewer’s assessment and hope these clarifications more accurately reflect the novelty and value of BlockSpec. As an early attempt at speculative decoding for diffusion LLMs—centered on low-confidence degeneration, trajectory-space draft generation, and inter-block parallelism—we believe our contributions are meaningful and foundational for future research.
> > >
> > > [1] DPad: Efficient Diffusion Language Models with Suffix Dropout

---

### Official Review · Reviewer_VSE3 · 2025-10-31

**Soundness:** 2
**Presentation:** 2
**Contribution:** 2
**Rating:** 4
**Confidence:** 3

**Summary:**

Common unmasking strategies, such as 'confidence threshold' and 'top-k', are slower on low-confidence tokens since only a single token can be sampled per step. The authors argue that low confidence occurs when dLLMs consider multiple possible trajectories. The paper, therefore, proposes a novel self-speculative decoding method called BlockSpec, which simultaneously searches for and verifies possible trajectories to select multiple tokens. BlockSpec achieves much higher throughput than the baseline and is twice as fast as concurrent work.

**Strengths:**

- BlockSpec is a customised, self-speculative decoding method designed specifically for dLLMs. It effectively addresses issues such as multi-token verification and changing the hidden state prefix.
- The paper also proposes a novel inter-block speculation method that substantially increases block efficiency.
- BlockSpec achieves a much higher TPS speedup compared to the baselines.
- BlockSpec is twice as fast as concurrent work.

**Weaknesses:**

- W1: There is no empirical evidence to support the argument that the low confidence tokens are due to plausible candidates. For example, the model would display low confidence if the problem was too complex for it to handle.
- W2: Since dLLMs use bidirectional attention, the hidden state of the prefix tokens varies as the model explores different nodes in the search tree. Therefore, the method is based on Fast-dLLM, which fixes the prefix token during the denoising process. However, since prefix caching leads to performance degradation, dependency on Fast-dLLM must be a weakness.
- W3: Despite its fast speed, BlockSpec shows similar performance to Fast-dLLM (dual cache). Formal Speculative Decoding methods maintain the original performance of the target models.
- W4: There has been no ablation study on the role of 'medium-confidence tokens'.
- W5: Short generation length. The generation lengths are fixed to 512, which is relatively short.

**Questions:**

- Typo in section 2 "When facing everal..."
- Is there any way to find an optimal tree configuration for arbitrary settings, such as model size, dataset, GPU FLOPS, and GPU memory bandwidth?
- W1: Could you show the results for more complex tasks, such as AIME?
- W3: Where does the performance degradation come from? I believe that the Fast-dLLM's performance would be the upper bound.

---

> ### Author Response · Authors · 2025-11-19
>
> Q1: Typo in section 2 "When facing everal..."
>
> Response to Q1: Thank you for catching this typo. We have corrected “everal” to “several” in Section 2 in the revised manuscript.
>
> Q2: Is there any way to find an optimal tree configuration for arbitrary settings, such as model size, dataset, GPU FLOPS, and GPU memory bandwidth?
>
> Response to Q2: Optimal tree configuration
> We thank the reviewer for raising this important question. It directly relates to how BlockSpec balances parallel efficiency and additional computation, and we appreciate the opportunity to clarify our design and search process.
>
> Direct answer：Our current work identifies tree configurations through extensive empirical comparison rather than a closed-form optimization rule. In particular, our search is guided by a practical observation from our experiments: BlockSpec typically reduces the total number of decoding iterations by 30–40%. Therefore, as long as the per-step additional latency introduced by exploring extra draft blocks stays below roughly 30%, the overall method still yields a general speedup. We would like to further explain our search strategies for tree structure in the aspect of serach space and search settings.
>
> Search space. We conducted extensive exploration over a broad space of draft-tree expansion strategies—including different branching factors, depth limits, duplicated vs.\ non-duplicated nodes, and symmetric vs.\ left-biased structures. Through systematic comparisons, we arrived at a hybrid design that provides the most robust trade-off under typical inference settings. The revised manuscript now includes a full discussion of these strategies in Appendix A.2, together with ablation results explaining why the hybrid configuration is preferred. While our current search is empirical, we think a automated search procedures could be possible for future work.
>
> Search settings. Instead of optimizing explicitly over FLOPS or memory bandwidth, we evaluate whether the single-step overhead stays within the acceptable range relative to the expected gain in parallel efficiency. This is precisely why we include the latency roofline analysis in Sec. 4.3, which directly visualizes how many additional blocks can be profitably explored before per-step latency outweighs the step-count reduction.
>
> From these measurements:
> -On A800, the per-step overhead remains acceptable for roughly 4 draft blocks (≈128 tokens).
> -On H800, the overhead stays small even up to around 9 draft blocks (≈288 tokens).
> These findings explain why W2D2 is preferred under A800 settings, while larger structures such as W3D3 become advantageous on stronger hardware.
>
> We also recognize that our explanation of compute overhead in initial submission was not sufficiently clear. Following the reviewer’s suggestion, we have significantly refined the description in Sec. 4.3, making explicit the relationship between additional blocks, per-step latency, and the resulting parallel decoding efficiency. Thanks again for highlighting this point, which allowed us to better articulate both the practical search strategy and the underlying trade-off governing BlockSpec.
>
> Q3: Could you show the results for more complex tasks, such as AIME?
>
> Response to Q3:
> We appreciate the reviewer’s suggestion. Following this comment, we conducted preliminary experiments on AIME using the same base models as in our main paper (LLaDA, Fast-dLLM, and our BlockSpec-accelerated variants), without adding complex CoT-style prompting or specific reasoning scaffolds. As a result, none of the models (neither the base diffusion LLMs nor our accelerated versions) were able to correctly solve a single AIME problem. We give a specific sample in a individual official comment below  for more discussion if interested.
>
> We believe this reveals an important characteristic of current discrete masked diffusion language models:
>  they struggle to perform long-horizon, multi-step mathematical reasoning, even before considering decoding strategies. This aligns with the absence of such benchmarks in prior diffusion-LLM work, and highlights a potential future direction for strengthening these models, e.g., via reasoning-oriented training or reinforcement learning.
>
> It is important to clarify that the focus of our work is on improving decoding efficiency for discrete diffusion LLMs, addressing limitations of applying speculative parallel decoding in this regime. Our method does not modify or enhance the underlying reasoning ability of the base models. The AIME results therefore reflect a limitation of the base models themselves rather than of BlockSpec.

---

> ### Author Response · Authors · 2025-11-19
>
> Q4: Where does the performance degradation come from? I believe that the Fast-dLLM's performance would be the upper bound.
>
> Response to Q4:
> We thank the reviewer for raising this important question. First, we clarify that our target model for speculative decoding is Fast-dLLM (Dual Cache), and all of our experiments are conducted on this baseline. Our initial submission did not emphasize this clearly, and we have added explicit statements in Sec. 4.1 (yellow-highlighted). We furthermore expanded Sec. 3.4 (Design of Suffix Tokens and KV Cache) to clarify how suffix [MASK] tokens and blockwise KV-cache mechanisms are handled in discrete diffusion LLMs.
>
> By comparing with Fast-dLLM (Dual Cache), our method achieves a comparable results with a small accuracy gap (<1%). The performance gap between Fast-dLLM (Dual Cache) with Fast-dLLM is mainly due to the use of blockwise cached attention mechanism: at the start of each block, all tokens perform one full-attention forward pass, and all prefix and suffix tokens' KV states are cached and remain fixed for the rest decoding steps of this block. This differs from full attention and introduces a small accuracy gap even before applying speculation.
>
> We observe two main sources of the slight accuracy differences(<1%) between our BlockSpec with Fast-dLLM (Dual Cache):
>
> - Higher parallelism changes diffusion trajectories. Unlike autoregressive speculative decoding—which is strictly lossless because generation is left-to-right and logits align exactly—discrete diffusion LLMs update multiple masked positions per step. Changing the number of tokens decoded per step (i.e., increasing parallelism) changes the trajectory of updated tokens, which can slightly alter intermediate logits. As visualized in Appendix A.8, BlockSpec’s more aggressive early-stage parallelism can cause small deviations at a few key tokens, leading to minor accuracy differences.
>
> - Inter-block speculation introduces additional blockwise attention and KV-cache discrepancies.
> Fast-dLLM (Dual Cache) caches KV states only within a single block. BlockSpec additionally performs inter-block blockwise cached attention, where drafts of next block attend to the shared root of the current block as we illustrated in Appendix A.3. This differs slightly from the full-attention behavior used during training, and introduces another small source of deviation.
>
> Across all datasets, both effects contribute only marginally, and the overall degradation remains <1%, which is within the range expected for discrete diffusion decoding under modified trajectories.
>
> We thank the reviewer again for prompting this clarification. In the revised manuscript, we added a concise explanation of these factors in the experimental analysis to provide a more complete discussion of accuracy differences.
>
> Q5：W1: There is no empirical evidence to support the argument that the low confidence tokens are due to plausible candidates. For example, the model would display low confidence if the problem was too complex for it to handle.
>
> Response to Q5:
> We thank the reviewer for this insightful comment. We fully agree that low-confidence tokens may arise from multiple causes, including (i) the presence of several plausible candidates and (ii) inherently complex problems where the model is uncertain regardless of candidate ambiguity.
>
> We would like to clarify that our method does not rely on low confidence being caused specifically by plausible alternatives. The mention of “plausible candidates” in the original submission was intended as a motivating example that inspired the design of our parallel speculative exploration. The core assumption behind BlockSpec is broader: even when a token has low confidence, it may still be correct, and verifying multiple possibilities in parallel can improve decoding efficiency without requiring confidence to be high.
>
> Importantly, this assumption remains valid under the reviewer’s “complex problem” scenario as well. When the model faces a difficult reasoning step and produces low-confidence candidates, speculative parallel verification is still beneficial: the correct token—if present—can be efficiently validated and accepted, leading to the same decoding speedup.
>
> We appreciate the reviewer for highlighting this point. In the revised version, if the reviewer doesn't mind, we have explicitly incorporated this broader interpretation of the causes of low confidence（including  plausible candidates and complex problem) and clarified that our assumption in Sec1 (highlighted in yellow ).

---

> ### Author Response · Authors · 2025-11-19
>
> Q6：W2: .. on Fast-dLLM must be a weakness?
>
> Response to Q6:
> We thank the reviewer for this insightful observation. Indeed, Fast-dLLM(Dual Cache) accelerates the original LLaDA model by introducing both prefix cache and suffix cache, effectively turning full attention at training time into a blockwise causal attention pattern at inference time. This train-test mismatch is empirically known to cause a small amount of performance degradation.
>
> However, we believe using such blockwise causal attention as the baseline remains meaningful and valuable for the following reasons:
>
> - The accuracy loss is minor and controlled.
> Both the Fast-dLLM paper and our experiments show that this attention gap leads to only about 1% accuracy drop, while bringing more than 7× speedup over the original LLaDA. In many practical settings, this trade-off between a minor loss in accuracy and a substantial gain in efficiency is acceptable. Also, the train-test mismatch can be further alleviated by updating the prefix and suffix KV caches more frequently, rather than only once at the beginning of each block as described in Sec.~3.4.
>
> - Blockwise causal models are an active and promising direction.
> Recent works have demonstrated that starting from a full-attention model and then applying blockwise causal attention with building blockwise causal variants of autoregressive models[1][2][3] can yield models that jointly optimize accuracy and speed. Our speculative decoding framework is fully compatible with these architectures and can be applied seamlessly to such blockwise causal diffusion or AR–diffusion hybrids.
>
> In this sense, the dependency on Fast-dLLM should be viewed not as a specific weakness of our method, but as a design choice aligned with a broader and increasingly popular family of blockwise causal architectures, to which BlockSpec can naturally extend.
>
> [1] Set Block Decoding is a Language Model Inference Accelerator
>
> [2] Sequential Diffusion Language Models
>
> [3] Fast-dLLM v2: Efficient Block-Diffusion LLM
>
> Q7： W3: Despite its fast speed, BlockSpec shows similar performance to Fast-dLLM (dual cache). Formal Speculative Decoding methods maintain the original performance of the target models.
>
> Response to Q7:
> Same with our Response to Q4
>
> Q8： W4: There has been no ablation study on the role of 'medium-confidence tokens'.
>
> Response to Q8:
> We thank the reviewer for pointing out this source of confusion. The phrase “medium-confidence tokens” in our original submission was a wording mistake. What we intended to describe are simply the top-k tokens among the low-confidence tokens, which in practice, often fall in a probability range around 0.3–0.6. These tokens were never meant to denote a distinct category separate from low-confidence tokens.
>
> We appreciate the reviewer for highlighting this ambiguity. In the revised manuscript, we have removed the term “medium-confidence tokens” and replaced it with a clearer description of top-k candidates within low-confidence positions (highlighted in green). We hope this clarification resolves the misunderstanding and more accurately reflects our method’s design.

---

> ### Author Response · Authors · 2025-11-19
>
> Q9: Short generation length. The generation lengths are fixed to 512, which is relatively short.
>
> Response to Q9:
> Thanks for raising this concern. In the initial submission, we fixed the generation length to 512 primarily to ensure a controlled and fair comparison across all evaluated decoding methods. This setting follows common practice in recent diffusion-LLM work and matches the evaluation protocols of Fast-dLLM, LLaDA, Dream, and other related models (typically evaluated at lengths 256 or 512).
>
> In the revised manuscript, we have added a new ablation study in Appendix A.9 evaluating BlockSpec and Fast-dLLM (Dual Cache) under longer generation lengths. Since our base model LLaDA is trained with a maximum context length of 4096 tokens (prompt + response), we tested generation lengths of 128, 256, 512, 1024, and 2048. Across all settings, BlockSpec consistently provides substantial throughput improvements over the baseline, demonstrating that the benefits of speculative parallel decoding are not restricted to short outputs. We present a quick result below. More explanation can be seen in Appendix A.9:
>
>
> Fast-dLLM (Dual Cache): 128 (1.5s), 256 (3.2s), 512 (6.6s), 1024 (10.2s), 2048 (22.4s)
>
> BlockSpec (W2D2):       128 (1.0s), 256 (2.3s), 512 (5.0s), 1024 (8.7s), 2048 (21.2s)
>
> Across all generation-length settings, BlockSpec consistently delivers meaningful inference speedup. We also observe that the relative improvement becomes smaller as the generation length increases (e.g., at 1024 and 2048 tokens). This behavior arises from the characteristics of current discrete diffusion masked models: after a certain number of blocks, the model typically predicts an [EOS] token, and all remaining positions are filled with placeholder [EOS] tokens. Once [EOS] is emitted, the decoding of each subsequent block requires only a single step, since the entire block is deterministically populated with [EOS].
>
> Because BlockSpec does not modify the model’s intrinsic denoising results, both BlockSpec and the baseline revert to a one-step-per-block pattern after [EOS] is generated. This naturally reduces the relative speedup advantage at longer sequence lengths, where a larger fraction of the decoding process occurs after [EOS]has already been produced.
>
> To illustrate this observation, Appendix~A.9 also visualizes the average number of decoding steps across blocks (with block size = 32). As shown in the figure, BlockSpec consistently requires fewer steps in the earlier blocks—where meaningful tokens are generated—while the latter blocks collapse to a single decoding step due to early \texttt{[EOS]} emission.
>
> We appreciate the reviewer for prompting this useful analysis, and we hope the expanded results more clearly illustrate how generation length interacts with speculative parallel decoding efficiency.

---

> ### Author Response · Authors · 2025-11-19
>
> Following is a demo response of our method's for a AIME question. The correct result is 116 and all of baseline models (llada, Fast-dLLMs) and our model get a wrong result.
>
> Queston: Please solve the following math problem. Requirements:
> 1. Think step by step.
> 2. Show each step of calculation and derivation in detail.
> 3. The final answer must be placed in \boxed{}
>
> Jen enters a lottery by picking $4$ distinct numbers from $S=\{1,2,3,\cdots,9,10\}.$ $4$ numbers are randomly chosen from $S.$ She wins a prize if at least two of her numbers were $2$ of the randomly chosen numbers, and wins the grand prize if all four of her numbers were the randomly chosen numbers. The probability of her winning the grand prize given that she won a prize is $\tfrac{m}{n}$ where $m$ and $n$ are relatively prime positive integers. Find $m+n$.
>
> Answer:
> To solve the problem, we need to determine the probability that Jen wins the grand prize given that she wins a prize. Let's break this down step by step.
>
> ### Step 1: Calculate the total number of ways to choose 4 numbers from \( S \)
>
> The set \( S \) has 10 elements. The number of ways to choose 4 numbers from 10 is given by the combination formula:
> \[
> \binom{10}{4} = \frac{10!}{4!(10-4)!} = \frac{10 \times 9 \times 8 \times 7}{4 \times 3 \times 2 \times 1} = 210
> \]
>
> ### Step 2: Calculate the number of ways to choose 4 numbers such that at least two of them are 2
>
> We need to count the number of ways to choose 4 numbers such that at least two of them are 2. This can be done by subtracting the number of ways to choose 4 numbers such that none of them are 2 from the total number of ways to choose 4 numbers.
>
> The number of ways to choose 4 numbers from the remaining 8 elements (since 2 is excluded) is:
> \[
> \binom{8}{4} = \frac{8!}{4!(8-4)!} = \frac{8 \times 7 \times 6 \times 5}{4 \times 3 \times 2 \times 1} = 70
> \]
>
> Thus, the number of ways to choose 4 numbers such that at least two of them are 2 is:
> \[
> 210 - 70 = 140
> \]
>
> ### Step 3: Calculate the probability that Jen wins the grand prize given that she wins a prize
>
> The probability that Jen wins the grand prize given that she wins a prize is the ratio of the number of favorable outcomes to the total number of outcomes:
> \[
> \frac{140}{210} = \frac{2}{3}
> \]
>
> ### Step 4: Find \( m + n \)
>
> Here, \( m = 2 \) and \( n = 3 \). Therefore, \( m + n = 2 + 3 = 5 \).
>
> ### Final Answer
>
> \boxed{5}

---

> ### Comment · Reviewer_VSE3 · 2025-11-28
>
> Thank you for the detailed replies. Most of my concerns are now addressed. I have one remaining question.
>
> Autoregressive speculative decoding is mathematically exact, but BlockSpec is not. I understand that the authors observe only small accuracy differences in the reported benchmarks. However, these benchmarks are not very difficult. This is why I asked about AIME. I accept that current diffusion LLMs cannot solve AIME, so experiments there are not meaningful.
>
> Instead, I would like to ask on a high level: in what situations do the authors expect BlockSpec to struggle? For example:
>
> * (1) On challenging reasoning tasks, could trajectory mismatch accumulate and cause larger performance declines?
> * (2) When low-confidence tokens appear repeatedly, could the speculative process become unstable?
> * (3) In long-horizon tasks, could inter-block speculation lead to additional errors?
>
> If the authors have any expectations or intuitions about these potential failure cases, it would help me better understand the robustness of the method.
>
> Thank you.

---

> > ### Author Response · Authors · 2025-12-02
> >
> > We sincerely thank the reviewer for the insightful follow-up question. This is an insightful point, and we are glad to further discuss the situations where BlockSpec may struggle. Our newly revised manuscript now includes an Exactness and Failure Analysis section in the Appendix, where we provide a formal exactness proof result under idealized conditions and an empirical failure case study. We summarize our intuitions and observations here.
> >
> > High-level intuition: Our analysis shows that BlockSpec is mathematically exact under an idealized setting—cached prefix/suffix KV states, no inter-block speculation, and deterministic top-1 decoding. Under this condition, BlockSpec and Fast-dLLM (dual cache) produce identical trajectories. Therefore, all discrepancies between the two methods arise solely from differences in the parallel decoding trajectories introduced by speculative lookahead. This provides a clear reference point for understanding potential failure cases.
> >
> > Empirical findings from Math (Levels 1–5). We evaluated BlockSpec across difficulty-separated Math tasks. As reported in the updated Appendix, BlockSpec and the baseline differ by only ~1% overall. However, failures concentrate on Level-4 problems rather than the easiest or the hardest ones. Our case study indicates these instances require both parallel decoding and mid-range multi-step reasoning—conditions under which speculative trajectories may diverge from those of the baseline.
> >
> > Replies to the reviewer’s three scenarios
> >
> > (1) Challenging long-horizon reasoning tasks. Interestingly, our experiments show that very hard (Level-5) problems do not incur significant accuracy degradation. The intuition is that when reasoning becomes highly demanding, both BlockSpec and the baseline naturally collapse to low parallelism (near single-token decoding), bringing BlockSpec closer to the mathematically exact regime. Thus, although trajectory mismatch can accumulate in principle, we do not observe pronounced degradation on the hardest tasks.
> >
> > (2) Repeated low-confidence tokens Repeated low-confidence positions do not destabilize BlockSpec.
> >  In fact, this is the regime where BlockSpec behaves most similarly to the baseline: when all positions exhibit low confidence, the tree becomes shallow; draft expansion collapses to top-1; the method reverts to near-exact behavior. Therefore, low-confidence streaks tend to reduce parallelism rather than introduce harmful deviations, and in our tests they do not lead to unstable decoding.
> >
> > (3) Long-horizon tasks with inter-block speculation. Inter-block speculation can introduce additional trajectory differences, but empirically the effect is mild. From our failure analysis, most errors stem from a small number of reasoning-critical tokens being decoded differently; these errors can originate from both intra-block and inter-block speculation. We do not observe systematic degradation attributable specifically to inter-block interactions.
> >
> > In short, BlockSpec tends to struggle not in fully hard reasoning tasks nor in low-confidence regions, but rather in medium-difficulty reasoning scenarios where:
> >
> > - parallel decoding remains active,
> >
> > - yet the reasoning chain is sensitive to perturbations in the trajectory.
> >
> > Finally, we would like to thank the reviewer again for raising this question. Our preliminary analysis suggests that such failure cases illuminate a fundamental trade-off between parallel decoding and maintaining stable reasoning chains in diffusion LLMs. We hope that these insights will help guide future research on more robust and principled speculative decoding mechanisms for dLLMs.

---

### Official Review · Reviewer_VDuL · 2025-10-31

**Soundness:** 2
**Presentation:** 2
**Contribution:** 3
**Rating:** 4
**Confidence:** 4

**Summary:**

The paper proposes Speculative Parallel Decoding (SPD) for diffusion-based LLMs, a topic gaining traction in the community. The method is training-free, maintains near-lossless performance on studied benchmarks, and introduces a simple draft tree generation algorithm with verification. The speed-up gains are good

**Strengths:**

- Proposes SPD for diffusion LLMs, which is an emerging and relevant area.
- Training-free approach, reducing complexity and resource requirements.
- Maintains almost lossless performance across evaluated benchmarks.
- Introduces a simple draft tree generation algorithm combined with verification.
- Works effectively with multi-token unmasking while maintaining accuracy.

**Weaknesses:**

- Draft tree generation algorithm lacks clarity in certain cases (e.g., W3D3(6) configuration).
- Missing citations for speculative decoding and diffusion-related prior work.
- Some results and configurations are not fully explained (e.g., latency trade-offs, inter-block setup).

Please see questions for more details

**Questions:**

Draft Tree Generation:
For W3D3(6), why is there no [1,4] node at step (or level) 2, given that the likelihood of [1,4] > [1,3,4] always?


Main Results:
Could the authors clarify that W2D2(3) was chosen because it offers lower latency compared to W3D3(6) on A800 early on in the results section?


Figure 5:
What does “base” refer to, and on which dataset are these results generated? Also, what is the difference between Fast-dLLM (dual cache) and the proposed method on H800, given that Fast-dLLM performs second best on A800?


Table 1:
HumanEval(0) shows BlockSpec outperforming baseline methods, while in other benchmarks BlockSpec performs slightly worse (as expected due to lack of lossless generation). Is there any justification for this anomaly?


Table 3:
Inter-block has the highest average tokens per step—what about its latency? Is the inter-block setup used only for 2 blocks? If extended to more than 2 blocks, does it start to hit TPS limits? Also, when doing inter-block, are draft trees for respective blocks generated independently?


Missing Citations:
Citations for speculative decoding and diffusion-based approaches are missing. This idea essentially extends speculative decoding for images as discussed in prior works.

- Diffusion Speculation


1. Diffusion Models are Secretly Exchangeable: Parallelizing DDPMs via Autospeculation
Hengyuan Hu, Aniket Das, Dorsa Sadigh, Nima Anari
arXiv:2505.03983


2. Accelerated Diffusion Models via Speculative Sampling
Valentin De Bortoli, Alexandre Galashov, Arthur Gretton, Arnaud Doucet
arXiv:2501.05370


- Speculative Decoding


1. Direct Alignment of Draft Model for Speculative Decoding with Chat-Fine-Tuned LLMs
Raghavv Goel, Mukul Gagrani, Wonseok Jeon, Junyoung Park, Mingu Lee, Christopher Lott
arXiv:2403.00858


2. Recursive Speculative Decoding: Accelerating LLM Inference via Sampling Without Replacement
Wonseok Jeon, Mukul Gagrani, Raghavv Goel, Junyoung Park, Mingu Lee, Christopher Lott
arXiv:2402.14160

---

> ### Author Response · Authors · 2025-11-19
>
> Q1. Draft Tree Generation — For W3D3(6), why is there no [1,4] ... ?
>
> Response to Q1:
> We sincerely appreciate the reviewer’s insightful question. Indeed, this observation captures the core motivation behind our tree-based trajectory exploration. Conceptually, our design initially follows the same intuition. Nevertheless, extensive experiments show that exploring more nodes after level 2 (e.g., keeping all top-k children) yields marginal gains in decoding acceptance rate while introducing substantial computational overhead, which may reduce overall speedup.
>
> In our final implementation, we empirically adopt a hybrid strategy: top-k expansion only at the root layer and  top-1 expansion for all deeper layers, as detailed in Appendix A.2. Under this hybrid strategy, the reviewer’s example is expected. When expanding from node [1], although draft [1,4] have a reasonably higher likelihood than [1,3,4], the model’s top-1 continuation from [1] happens to be [1,3]. Therefore, the node [1,4] is intentionally pruned—not because its likelihood is low or invalid, but because deeper layers never include more than one child.
>
> We especially thank the reviewer for raising this point and this provides a valuable opportunity for us to expose more of our design rationale and empirical analysis. Due to page-limit and clarity considerations, these design choices and ablation studies were omitted from the initial submission. In the revised version, we added these details to Section 3.2(highlighted in yellow) and a full Appendix A.2, where we explicitly compare different tree expansion strategies.
>
> Q2. Main Results — Could the authors clarify that W2D2(3)...?
>
> Response for Q2:
> We sincerely thank the reviewer for this helpful suggestion. In the revised manuscript, we have added an explicit clarification in Sec. 4.1 (highlighted in blue) that W2D2(3) is chosen for A800 evaluations because it achieves the best latency–efficiency balance among all tested configurations.
>
> In addition, we would like to explicitly highlight the underlying trade-off. While larger draft-tree structures such as W3D3(6) provide higher acceptance rates and reduce decoding iterations, they also incur heavier per-step verification and expansion overhead, which can outweigh the benefits and lead to worse end-to-end latency on A800. To make this trade-off clearer, we have revised the total Sec. 4.3, and explain how this trade-off inspire us to do experiments on higher-compute hardware (H800) in Sec.4.4 to demonstrate the potential of larger tree structures when more compute is available.
>
> Q3. Figure 5 — What does “base” refer to, and on which dataset ...?
>
> Response for Q3:
> Thanks for pointing out this ambiguity. In Figure 5, “base” refers to Fast-dLLM (Dual Cache), which is the baseline used for all BlockSpec experiments. The results in the figure are generated on the HumanEval benchmark using LLaDA-7B-Instruct.
>
> Due to page-limit, our initial submission did not explicitly restate this baseline choice in the figure caption. In the revised version, we have clarified this in both Sec. 4.1(yellow-highlighted) and Sec. 4.4 (yellow-highlighted), explicitly noting that Fast-dLLM (Dual Cache) is used as the baseline throughout all latency and throughput analysis.

---

> ### Author Response · Authors · 2025-11-19
>
> Q4. Table 1: HumanEval(0) shows BlockSpec outperforming baseline methods, while in other benchmarks BlockSpec performs slightly worse ...?
>
> Response for Q4:
> We appreciate the reviewer for raising this observation. In general, BlockSpec achieves higher parallel decoding efficiency by exploring a more aggressive decoding trajectories. This may occasionally introduce small accuracy drops—consistent with our results on GSM8K, MATH, and MBPP.
>
> However, HumanEval behaves differently from the other benchmarks. We observed that diffusion LLMs tend to exhibit high output variance on HumanEval—even across different devices platform. Moreover, we also find that enabling KV-cache consistently improves performance on HumanEval, a trend that is aligned with the original Fast-dLLM paper. We believe that the distinct behavior on HumanEval largely stems from the small size and inherent noisiness of the dataset for diffusion-based models. In addition, the tasks in HumanEval are relatively simple, meaning that higher parallelism can produce more aggressive yet simpler decoding trajectories that may occasionally lead to better program-level correctness.
>
> Fast-dLLM, cuda 12.3, 12.5, 12.7: 43.3
>
> Fast-dLLM, cuda 12.2: 44.5
>
> Fast-dLLM, cuda 12.8:  40.9
>
> Fast-dLLM (dual cache): 45.7
>
> In our further case study (visualizations included in the updated Appendix A.8), we found concrete cases where BlockSpec’s more aggressive path corrected a token that the baseline mispredicted, resulting in a correct result—while in other datasets such isolated tokens do not have such high impact. This sensitivity explains why BlockSpec is slightly better on HumanEval despite being generally more “speculative” than Fast-dLLM.
>
> Q5: Table 3: Inter-block has the highest average tokens per step—what about its latency? Is the inter-block setup used only for 2 blocks? If extended to more than 2 blocks, does it start to hit TPS limits? Also, when doing inter-block, are draft trees for respective blocks generated independently?
>
> Response for Q5:
> Our direct answers are as follows: (1) Latency. We have added a dedicated analysis of inter-block latency in Sec. 4.3 and Appendix A.3, showing how higher tokens-per-step interacts with the additional verification cost.(2) Yes. Inter-block speculation is used only between two consecutive blocks. This design avoids the rapidly increasing verification overhead that would occur when extending to more blocks. We clarify this in Sec. 3.4 and provide visual explanations in Appendix A.3. (3) Draft-tree construction. Yes, the draft trees for the two blocks are generated independently, with details, examples, and visualizations added in Appendix A.3.
>
> We thank the reviewer again for highlighting this. Their questions provided a valuable opportunity for us to clarify and more fully articulate the design of inter-block speculation. Motivated by these comments, we have substantially expanded the corresponding sections in Appendix A.3 in the revised manuscript. Appendix A.3 now includes additional design motivation, detailed visualizations of the inter-block workflow, attention-mask diagrams, pseudocode, and extended experimental comparisons. And we hope this revised presentation give readers a clearer and more complete understanding of the design of inter-block speculation.
>
> Q6：Missing Citations: Citations for speculative decoding and diffusion-based approaches are missing. This idea essentially extends speculative decoding for images as discussed in prior works.
>
> Response for Q6: Thanks for pointing out these references. We have added these suggested citations to the revised manuscript. We also would like to clarify that our work specifically targets discrete masked diffusion language models, whereas the cited speculative-decoding methods for diffusion mainly focus on continuous denoising diffusion models. Although related in high-level motivation, the underlying formulations and verification mechanisms differ substantially. The omission in the paper was due to page-limit constraints, and we appreciate the reviewer’s reminder, which helped us complete and broaden the contextualization of our contributions.

---

### Author Response · Authors · 2025-12-02
**General Response (part 1)**

We sincerely thank all reviewers for their thoughtful and constructive feedback. We are glad that several key strengths of the paper were consistently recognized, including:

- A training-free speculative decoding framework tailored for diffusion LLMs;

- Substantial decoding acceleration (7–14× over vanilla dLLMs and ~30% latency reduction over Fast-dLLM (Dual Cache), reduces the number of denoising iterations by 30–40%, and increases effective tokens-per-step from 2.6 to 4.3);

- A tree-structured parallel verification mechanism that operates under bidirectional attention;

- A new inter-block speculation strategy enabling segmented lookahead beyond single-block decoding;

- Clear experimental validation across models (LLaDA, Fast-dLLM), datasets, and hardware (A800/H800).

- Improved clarity and presentation in the revision. The updated manuscript now includes a formal exactness theorem, expanded failure-case analysis, precise attention-mask diagrams, clearer prefix/suffix KV caching explanations, and expanded presentation of interblock speculation in Appendix.

Below we address the main cross-reviewer concerns and summarize the clarifications and revisions incorporated into the updated manuscript.

## 1. Novelty and Contribution
We appreciate that most reviewers recognized the contributions of BlockSpec. In response to requests for additional clarification, we have refined the presentation to more explicitly distinguish BlockSpec from both Fast-dLLM (Dual Cache) and existing autoregressive speculative decoding methods.
While Fast-dLLM (Dual Cache) provides a systems-level optimization—reducing per-step FLOPs through blockwise KV caching—BlockSpec introduces a complementary and algorithmic-level speculative decoding framework designed specifically for diffusion LLMs. This framework enables parallel multi-token denoising under bidirectional attention, a setting fundamentally different from AR models.
The key innovations, now clarified more explicitly in the revision, include:

- Blockwise speculative decoding for arbitrary-order denoising. Unlike AR speculative decoding, dLLM decoding operates over masked multi-token trajectories, requiring a redesign of draft construction, trajectory ordering, and correctness criteria under bidirectional attention.
- A principled analysis and remedy for low-confidence degeneration. We identify the collapse of parallelism under low confidence as a unique bottleneck in diffusion LLMs and design draft collection & verification specifically targeting this issue.
- A carefully explored tree-structured draft scheme adapted to dLLM verification cost. Multiple expansion policies (top-k at root, top-1 at deeper layers, duplication control, depth–width trade-offs) were empirically explored and distilled into a hybrid design (Appendix A.2).
- Novel Inter-block speculation in dLLMs. Enabled by bidirectional visibility, segmented parallel decoding across two blocks has no AR analogue and required new attention-mask design, verification logic, and correctness analysis.
To make these distinctions clear, we will add a dedicated paragraph(as we shown in comment of Reviewer DaQs) in the revision explicitly contrasting BlockSpec with Eagle/Medusa-style AR speculative decoding.

## 2. Performance Improvements
Most reviewers recognized the strong empirical performance of BlockSpec, particularly its gains in decoding speed and parallel efficiency. We appreciate these comments and emphasize that BlockSpec delivers meaningful improvements even on top of an already optimized baseline (Fast-dLLM with Dual Cache).

Concretely, BlockSpec achieves:

- ≈40% fewer denoising iterations,

- ≈65% higher tokens-per-step (from 2.6 → 4.3),

- ≈30% additional end-to-end latency reduction beyond Dual Cache.

These results demonstrate that our method scales parallelism effectively without sacrificing accuracy (<1% difference). Importantly, this improvement is orthogonal to Dual Cache:

- Dual Cache reduces per-step FLOPs,

- BlockSpec reduces the number of required steps.

Because the two components optimize different parts of the decoding pipeline, their benefits accumulate cleanly, making BlockSpec’s effect clearly visible even on an already accelerated system.
We also highlight that, compared with concurrent speculative-decoding work for diffusion LLMs (e.g., dPAD, Spiffy), BlockSpec provides a complementary and practically meaningful speedup, particularly for low-confidence and mid-depth denoising stages where existing methods remain slow. Reviewers widely acknowledged these improvements as a key strength of our submission.

---

> ### Author Response · Authors · 2025-12-02
> **General Response (part 2)**
>
> ## 3. Clarification of blockwise masks, suffix tokens, and KV caching
> Multiple reviewers give concerns about Fig. 3(3) and the masking / visibility rules. We agree the earlier figure was misleading in the initial submission. In the revised manuscript, we have:
>
> - redrew all attention-mask diagrams (Sec3.1, Appendix A.3, )
>
> - clarified how prefix/suffix KV states are recomputed and cached at each block (Sec 3.4)
>
> - explained draft-token visibility and isolation across branches
>
> - redesigned the terminology to avoid conflation with causal attention
>
> - added full descriptions of inter-block masking (Appendix A.3)
>
> These additions resolve most of the reviewers' concerns.
>
> ## 4.  Better revisions prompted by reviewers
> We sincerely appreciate the reviewers’ constructive dialogue throughout the rebuttal process. Their thoughtful questions and clarifications directly helped us refine both the presentation and empirical analysis, and the paper is significantly stronger as a result.
>
> In response to the collective feedback, we have incorporated the following improvements:
>
> - Added complete attention-mask diagrams for both intra-block and inter-block speculation.
>
> - Expanded Appendix A.2 with detailed tree strategies and ablation results.
>
> - Expanded Appendix A.3 with inter-block design discussions, diagrams, and pseudocode.
>
> - Added long-sequence decoding experiments up to 2048 tokens (Appendix A.9).
>
> - Clarified the baseline definition, dataset configurations, and figure captions (Sec. 4.1, Fig. 5).
>
> - Added missing citations (Autospeculation, AccelDiffusion, AR speculative decoding, etc.).
>
> - Improved terminology (e.g., “medium-confidence tokens”, “degradation”) for clarity.
>
> - Fixed typos and enhanced overall figure readability.
>
> - Formal exactness theorem proof and failure case study (Appendix A.10)
>
> We are grateful to the reviewers—their deep and thoughtful engagement significantly strengthened the final submission, and many of the improvements in the revised paper were made possible through this productive and collegial dialogue.
>
> ## Closing remarks
> We are deeply grateful for the reviewers’ thoughtful engagement and the constructive discussions throughout the review process. The detailed questions and insightful feedback greatly helped us refine conceptual framing, strengthen the empirical analysis, and significantly improve the clarity of the paper.
>
> BlockSpec represents one of the earliest explorations of speculative decoding for diffusion LLMs—a setting fundamentally different from autoregressive models. As an initial step, our study sheds light on:
>
> - how low-confidence degeneration emerges in dLLMs,
>
> - how speculative trajectories interact with bidirectional attention,
>
> - how blockwise and cross-block parallelism can be effectively realized, and
>
> - how near-lossless decoding remains achievable even under semi-autoregressive constraints.
>
> We hope that the insights, analyses, and design principles consolidated through this review process will contribute to future work on more principled and robust speculative decoding techniques for diffusion LLMs. We sincerely thank the reviewers again for their invaluable input and for helping us deliver a much stronger final submission.

---

### Meta-Review · Area_Chair_kQ1i · 2026-01-07

**Summary:**

Reviewers raised concerns primarily around the conceptual novelty of the proposed blockwise speculative decoding framework relative to existing speculative decoding and diffusion decoding strategies. In particular, several reviewers questioned whether the proposed tree-based trajectory exploration constitutes a fundamentally new decoding paradigm or a heuristic extension of prior approaches. Additional concerns were raised regarding the clarity of the method description, the strength of theoretical justification, and whether the empirical speedups generalize across models, tasks, and decoding settings beyond the presented benchmarks.

**Reviewer Concerns:**

The rebuttal helped clarify implementation details, decoding procedures, and experimental settings, and partially addressed questions about how BlockSpec differs operationally from prior speculative decoding methods. The authors also provided reasonable explanations for some design choices and acknowledged limitations candidly. However, the core concerns regarding conceptual novelty, the lack of strong theoretical grounding, and insufficient evidence that the gains are robust and broadly applicable remain largely unresolved. In particular, the rebuttal did not convincingly demonstrate that the proposed approach represents a principled advance rather than an engineering-driven heuristic tailored to specific diffusion decoding setups.

**Reviewer Scores:**

4446. Had reviewers been able to participate fully in post-rebuttal discussion, some might have slightly increased their scores to reflect improved clarity and responsiveness from the authors. However, it is unlikely that these changes would have been sufficient to cross the acceptance threshold, as the main concerns about novelty and generality would likely have persisted.

---

### Decision · Program_Chairs · 2026-01-26

Reject